# Let LLMs Speak Embedding Languages: Generative Text Embeddings via Iterative Contrastive Refinement

**Yu-Che Tsai**[*,1]    **Kuan-Yu Chen**[*,1]    **Yuan-Chi Li**[1]

**Yuan-Hao Chen**[1]    **Ching-Yu Tsai**[1]    **Shou-De Lin**[1,2]

[1]Department of Computer Science and Information Engineering, National Taiwan University
[2]National Taiwan University AI Center of Research Excellence
Taipei, Taiwan
{f09922081,d13922034,b09902110,b11902172,r14922006,sdlin}@csie.ntu.edu.tw

## Abstract

Existing large language model (LLM)-based embeddings typically adopt an encoder-only paradigm, treating LLMs as static feature extractors and overlooking their core generative strengths. We introduce GIRCSE (Generative Iterative Refinement for Contrastive Sentence Embeddings), a novel framework that leverages autoregressive generation to iteratively refine semantic representations. By producing sequences of soft tokens optimized under contrastive objective, GIRCSE captures latent concepts and implicit semantics that encoder-only methods often miss. To guide this process, we propose an Iterative Contrastive Refinement (ICR) objective that encourages each refinement step to yield better representations. Extensive experiments show that GIRCSE outperforms strong LLM-based embedding baselines on the MTEB benchmark and instruction-following tasks. Moreover, GIRCSE exhibits an emergent test-time scaling property: generating more tokens at inference steadily improves embedding quality. Our results establish generative iterative refinement as a new paradigm for representation learning. Our code and pre-trained models are available at `https://github.com/Roytsai27/GIRCSE`.

## 1 Introduction

Text embeddings are fundamental to a wide range of natural language processing (NLP) applications, including information retrieval, semantic search, clustering, and recommendation (Karpukhin et al., 2020; Liu et al., 2024; Xiong et al., 2021). With the rise of large language models (LLMs), representation learning has advanced further: fine-tuning LLMs on large corpora now yields superior performance on several embedding benchmarks (Tao et al., 2024).

However, current LLM-based embeddings typically operate as single-pass feature extractors: embeddings are extracted in a single forward pass with contrastive learning objectives, without leveraging the generative capacity of LLMs. This overlooks a key strength of pretrained LLMs—their ability to reason and iteratively refine through autoregressive generation (Wei et al., 2022; Muennighoff et al., 2025). This raises a fundamental question: *Can LLM-based embedding models also benefit from iterative generation?* We hypothesize that generation enables iterative refinement of embeddings, allowing models to progressively consolidate semantics over multiple steps rather than encoding all semantics in a single pass.

**Challenges.** Designing effective generative embeddings presents several challenges. First, naive generation degrades embedding quality since pretrained LLMs are optimized for fluent text, not tokens aligned with semantic similarity (see Section 5.1). Second, unlike traditional language modeling, there is no clear generation target: it is unclear what content the model should generate to obtain universally useful embeddings. Third, existing embedding learning frameworks do not ac-

---

[*]Equal contribution.

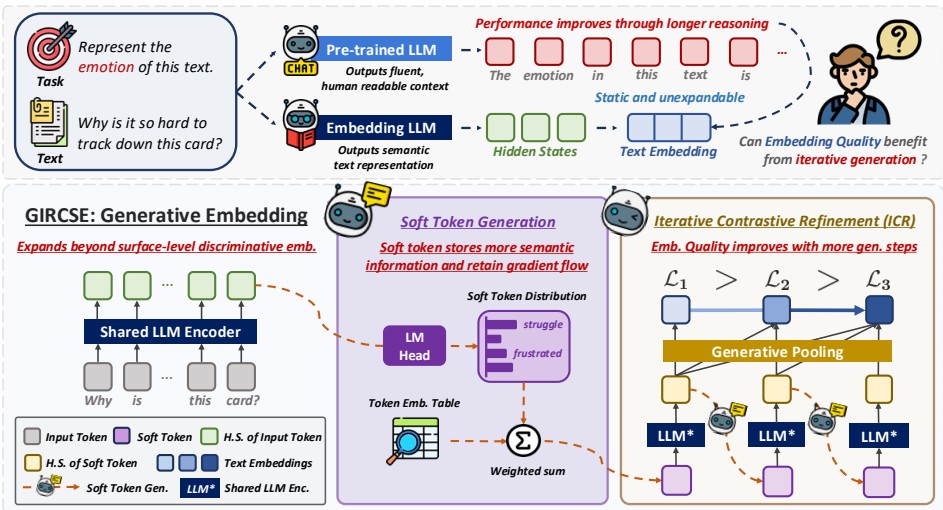

Figure 1: **Top:** Comparison between embedding LLMs that extract static representations and generative LLMs that can iteratively refine through reasoning. **Bottom:** Overview of GIRCSE. Our framework combines Soft Token Generation and Iterative Contrastive Refinement to enable end-to-end generative training.

commodate multi-step generative refinement. Therefore, it is necessary to develop new training paradigms that provide meaningful supervision for generative embeddings.

**Motivation.** We argue that LLMs should learn to *speak an embedding language*: generating tokens not constrained by human readability but optimized for semantic representation. Crucially, these tokens should be discovered through **end-to-end training jointly with contrastive objectives**, enabling the model to generate semantically meaningful tokens for iterative embedding refinement.

Building on this motivation, we propose **GIRCSE**—*Generative Iterative Refinement for Contrastive Sentence Embeddings*—a novel framework that bridges this gap between generative LLM capabilities and embedding optimization. GIRCSE consists of two major innovations: (1) **Soft Token Generation** preserves differentiability for end-to-end contrastive training and captures richer semantics by retaining the diversity of the full probability distribution. (2) **Iterative Contrastive Refinement (ICR)** provides contrastive supervision at every generation step, forcing the early generated tokens to capture useful semantics while later tokens progressively refine representations. As illustrated in Fig. 1 and detailed in Section 5.2, this paradigm enables GIRCSE to generate instruction-aware refinement tokens (e.g.,"frustrated" and "struggle") that capture the implicit emotion beyond the surface text. In summary, we make the following contributions:

- **Novel embedding framework.** We propose **GIRCSE**, a novel end-to-end framework that integrates autoregressive generation with contrastive learning. Unlike prior methods, GIRCSE generates *soft refinement tokens* without explicit targets, progressively distilling semantics into high-quality embeddings.

- **Superior performance.** We compare GIRCSE with 18 state-of-the-art embedding models. By generating only up to 5–20 additional tokens, GIRCSE ranks within top 5–6 on MTEB and top 2–3 on instruction following, leading to the best overall ranking across benchmarks. Meanwhile, GIRCSE consistently shows stable improvements over reproduced fair baselines on different backbone and varying data scales.

- **Test-time scaling ability for text embedding.** We demonstrate that GIRCSE exhibits consistent embedding quality improvements with increased refinement steps at inference time, representing a novel scaling paradigm for embedding models analogous to test-time compute scaling in reasoning LLMs.

## 2 RELATED WORK

**Early Embedding Models.** Text embedding methods have evolved from traditional word-level representations to sophisticated neural approaches. Early methods like Word2Vec (Mikolov et al., 2013) and GloVe (Pennington et al., 2014) captured basic semantic relationships but lacked contextual understanding. The introduction of transformer-based models marked a significant advancement, with BERT-based approaches like Sentence-BERT (Reimers & Gurevych, 2019) and SimCSE (Gao et al., 2021) establishing contrastive learning as the dominant paradigm for learning sentence representations.

**LLM-based Embedding models.** Recent works have successfully adapted LLMs for representation learning through various architectural and training modifications. E5-mistral (Wang et al., 2024) is one of the early works that demonstrated that fine-tuning LLM could significantly outperform early stage methods. Recognizing that the unidirectional attention mechanism in LLMs may limit text embedding quality, LLM2Vec (BehnamGhader et al., 2024) introduces a bidirectional attention mechanism combined with average pooling to enhance embedding quality. NV-Embed (Lee et al., 2024) further improves the pooling method by incorporating an additional Latent Attention Layer and implements a two-stage training strategy to address the challenge of false negatives in non-retrieval tasks. BGE-en-icl (Li et al., 2025) suggests that retaining the original framework of LLMs and leveraging in-context learning is the optimal approach for generating text embeddings.

**Towards Generative Text Embedding.** A smaller line of research has explored generative approaches for text embeddings. For example, Inbedder (Peng et al., 2024) combines instruction finetuning with token generation, achieving strong performance on instruction-following tasks but showing limited generalization to broader tasks (see Table 2). As summarized in Table 1, most existing approaches differ only in pooling strategies or auxiliary training techniques, while generative embeddings remain largely underexplored.

Table 1: Comparison of LLM-based embedding models. "Bidir." indicates bidirectional attention, "GP" means generated tokens pooling, "TTS" refers to test-time scaling capability.

| Method | Attention | Pooling | LoRA | Generation | TTS | Gen. Token | Training Obj. |
|---|---|---|---|---|---|---|---|
| E5-MISTRAL | Causal | EOS | ✓ | ✗ | ✗ | N/A | sup. CL |
| SFR-EMBEDDING | Causal | EOS | ✗ | ✗ | ✗ | N/A | sup. CL |
| BGE-EN-ICL | Causal | EOS | ✓ | ✗ | ✗ | N/A | ICL & sup. CL |
| GRITLM | Bidir. | Avg. | ✗ | ✗ | ✗ | N/A | NTP & sup. CL |
| LLM2VEC | Bidir. | Avg. | ✓ | ✗ | ✗ | N/A | MLM & sup. CL |
| GTE-QWEN2 | Bidir. | Avg. | ✗ | ✗ | ✗ | N/A | unsup. CL & sup. CL |
| NV-EMBED-V1 | Bidir. | LAT | ✓ | ✗ | ✗ | N/A | Two-stage sup. CL |
| INBEDDER | Causal | GP | ✓ | ✓ | ✗ | Hard Token | Instruction Tuning |
| **GIRCSE** | Causal | GP | ✓ | ✓ | ✓ | **Soft Token** | sup. CL & **ICR** |

## 3 GIRCSE: FROM DISCRIMINATIVE TO GENERATIVE EMBEDDING

We now detail our proposed generative embedding framework. Section 3.1 first establishes our core autoregressive embedding generation process, introducing the fundamental concepts and notation. Section 3.2 then details the soft token generation mechanism that enables differentiable optimization within this framework. Finally, Section 3.3 presents our iterative contrastive refinement objective, guiding the model towards progressively higher-quality representations.

### 3.1 GENERATIVE EMBEDDING FRAMEWORK

We consider a pretrained language model with parameters $\psi = \{\mathbf{E}, \theta, \phi\}$, where $\mathbf{E} \in \mathbb{R}^{|\mathcal{V}| \times d}$ is the token embedding matrix, $\theta$ denotes the parameters of the Transformer decoder, and $\phi$ corresponds to the parameters of the LM head for next token generation. Here $d$ is the embedding dimension and $|\mathcal{V}|$ is the vocabulary size. Given an input sequence of $N$ tokens $\mathbf{T} = \{t_1, t_2, \ldots, t_N\}$ from vocabulary $\mathcal{V}$, we first obtain token embeddings $\mathbf{X}$ as:

$$\mathbf{X} = (\mathbf{E}[t_1], \mathbf{E}[t_2], \ldots, \mathbf{E}[t_N]) \in \mathbb{R}^{N \times d}. \tag{1}$$

Next, our goal is to autoregressively generate a sequence of $K$ auxiliary soft tokens $\mathbf{S} = (\mathbf{s}_1, \mathbf{s}_2, \ldots, \mathbf{s}_K) \in \mathbb{R}^{K \times |\mathcal{V}|}$ that iteratively refines the representation space. Each soft token $\mathbf{s}_k$

is generated autoregressively conditioned on the input sequence and previously generated tokens:

$$p_\psi(\mathbf{D} \mid \mathbf{T}) = \prod_{k=1}^{K} p_\psi(\mathbf{d}_k \mid \mathbf{T}, \mathbf{D}_{<k}) \tag{2}$$

where $p_\psi$ represents the generative distribution and $\mathbf{S}_{<k} = (\mathbf{s}_1, \ldots, \mathbf{s}_{k-1})$ are the previously generated soft tokens. The soft tokens are then mapped into embedding space[1], producing $\mathbf{D} = (\mathbf{d}_1, \ldots, \mathbf{d}_k) \in \mathbb{R}^{K \times d}$, and is concatenated with input embeddings $\mathbf{X}$ to further feed into the Transformer decoder $f_\theta$:

$$\mathbf{H} = f_\theta([\mathbf{X}; \mathbf{D}]) = (\mathbf{h}_1^{(L)}, \mathbf{h}_2^{(L)}, \ldots, \mathbf{h}_{N+K}^{(L)}) \in \mathbb{R}^{(N+K) \times d}, \tag{3}$$

where $\mathbf{h}_i^{(L)}$ denotes the hidden state of the $i$-th token at the final (i.e., $L$-th) layer. We then extract the representations corresponding to the generated soft tokens, and aggregate them into a single representation $\mathbf{z}$ via a pooling operation:

$$\mathbf{z} = \mathcal{P}(\mathbf{G}) = \frac{1}{K} \sum_{i=1}^{K} \mathbf{g}_i, \quad \mathbf{G} = (\mathbf{g}_1, \ldots, \mathbf{g}_K) = (\mathbf{h}_{N+1}^{(L)}, \ldots, \mathbf{h}_{N+K}^{(L)}) \in \mathbb{R}^{K \times d}, \tag{4}$$

where $\mathcal{P}$ denotes a general pooling function, with mean pooling as our default choice.

**Computational Considerations.** Since our approach involves iterative generation with $K$ steps, it naturally incurs a higher computational cost compared to single-step baselines (Appendix E). However, we find that generating only a small number of tokens (e.g., $K = 5$ or 10) is sufficient to achieve strong performance (Section 5.1). Moreover, this cost could be largely mitigated via KV caching (Appendix D), where the FLOPs are reduced to nearly the same level as standard embedding models (within $\sim 1.0$–$1.1\times$).

## 3.2 SOFT TOKEN GENERATION

A critical challenge in our autoregressive framework is to maintain the differentiability throughout the generation process during training. Traditional discrete token sampling would break gradient flow, preventing end-to-end optimization. We address this through a novel soft token generation mechanism that preserves continuous optimization while capturing rich semantic information.

At each generation step $k \in \{1, \ldots, K\}$, the generative distribution $p_\psi$ is instantiated via the LM head $\phi$. Let $\mathbf{h}'_{k-1} = \mathbf{h}_{N+k-1}^{(L)}$ denote the last layer hidden representation produced by the encoder given the input sequence and the previously generated soft tokens up to step $k - 1$, the LM head then produces a soft token $\mathbf{s}_k \in \mathbb{R}^{|\mathcal{V}|}$ as a probability distribution over the vocabulary:

$$\mathbf{s}_k = \mathrm{softmax}(\mathbf{W}_\phi \mathbf{h}'_{k-1} + \mathbf{b}_\phi), \tag{5}$$

where $\mathbf{W}_\phi \in \mathbb{R}^{|\mathcal{V}| \times d}$ is the LM head weight matrix and $\mathbf{b}_\phi$ is the bias term. Given the soft token $\mathbf{s}_k$, its embedding $\mathbf{d}_k \in \mathbb{R}^d$ is obtained by computing a convex combination of all token embeddings according to their predicted probabilities:

$$\mathbf{d}_k = \sum_{i=1}^{|\mathcal{V}|} s_{k,i} \mathbf{e}_i, \tag{6}$$

where $s_{k,i}$ is the $i$-th component of $\mathbf{s}_k$ and $\mathbf{e}_i$ is the $i$-th row of the embedding matrix $\mathbf{E}$. This soft token generation approach offers two advantages: (1) **Differentiability**: The weighted combination preserves gradients throughout the generation process, enabling end-to-end training with contrastive objectives. (2) **Semantic Richness**: Rather than collapse the next-token distribution into a single token, soft tokens capture the semantic diversity of the full probability distribution.

---

[1]We defer the detailed soft token generation mechanism to Section 3.2, while here we focus on the overall framework

---

**Algorithm 1:** GIRCSE: Autoregressive Generative Embedding

---

**Input:** Input tokens $(t_1, \ldots, t_N)$, generation steps $K$, embedding matrix $\mathbf{E}$, LLM decoder $f_\theta$,
      LM head parameters $W_\phi$ and $b_\phi$, pooling function $\mathcal{P}$
**Output:** Final embedding $\mathbf{z} \in \mathbb{R}^d$

```
X ← (E[t₁],...,E[t_N])                        // Embed the input tokens
D ← []                  // Initialize the list of generated embeddings
for k = 1 to K do
    H ← f_θ([X;D])          // Forward with input and generated tokens
    h'_{k-1} ← H[N + k − 1]         // Last hidden state for next-token
     prediction
    s_k ← softmax(W_φ h'_{k−1} + b_φ)      // Compute soft token distribution
    d_k ← Σ^{|V|}_{i=1} s_{k,i} e_i               // Compute soft embedding
    D ← D‖d_k                    // Append soft embedding for next step
H ← f_θ([X;D])
G ← (H[N + 1],...,H[N + K])        // Collect the last K hidden states
z ← P(G)                // Pool generated representations into a single
 embedding
return z
```

---

## 3.3 Guiding Generative Embedding with Iterative Contrastive Refinement

To guide the generative embedding process towards high-quality representations, we introduce an *iterative contrastive refinement (ICR)* objective that encourages each generation step to yield increasingly refined representations. ICR combines (1) **Stepwise Contrastive Loss**, which supervises each generation step with contrastive loss, and (2) **Iterative Refinement Regularization**, which enforces progressive embedding quality improvement for each step.

**Stepwise Contrastive Loss.** In autoregressive soft token generation, supervising only the final embedding (i.e., $K$-th generation step) might collapse intermediate steps into trivial or noisy representations. We instead apply contrastive supervision at every generation step. Concretely, for step $k$, we pool the first $k$ generated tokens to form an intermediate embedding $\mathbf{z}_k = \mathcal{P}(\mathbf{G}_{1:k})$ following Eq. (4). Given a query–document pair $(q, d^+)$, we compute the contrastive loss for all generation steps as:

$$\mathcal{L}_{\text{contrast}} = \sum_{k=1}^{K} \mathcal{L}_k, \quad \mathcal{L}_k = -\log \frac{\exp\big(\sigma(\mathbf{z}_k^q, \mathbf{z}_k^{d^+})/\tau\big)}{\sum_{d \in \mathcal{B}} \exp\big(\sigma(\mathbf{z}_k^q, \mathbf{z}_k^d)/\tau\big)}, \tag{7}$$

where $\mathcal{B}$ denotes the document set (both positive and negative documents), $\sigma$ is the cosine similarity function, and $\tau$ is the temperature hyperparameter. This stepwise supervision ensures all intermediate representations align with the contrastive objective, preventing early steps from drifting and providing richer supervision.

**Iterative Refinement Regularization.** We empirically observe that simply increasing the number of generation steps does not guarantee improved embedding quality, as LLMs often produce highly similar tokens which leads to redundant information in the multi-step process. To address this, we introduce a regularization term that encourages monotonic improvement across generation steps:

$$\mathcal{L}_{\text{reg}} = \frac{1}{K-1} \sum_{k=1}^{K-1} \max\big(\log \mathcal{L}_{k+1} - \log \mathcal{L}_k, \, 0\big). \tag{8}$$

This regularization term penalizes cases where later generation steps fail to outperform earlier ones. Finally, the overall fine-tuning objective for generative embeddings combines the two terms: $\mathcal{L}_{\text{total}} = \mathcal{L}_{\text{contrast}} + \lambda \mathcal{L}_{\text{reg}}$, where $\lambda$ is a hyperparameter that balances contrastive alignment and refinement regularization.

We summarize the overall training procedure in Algorithm 1. During each training iteration, we sample a mini-batch of queries and documents. For every input sequence, the model performs autoregressive soft token generation for $K$ steps to obtain the sequence of embeddings. Finally, the

model parameters $\psi$ are updated end-to-end via gradient descent to minimize the combined objective $\mathcal{L}_{\text{total}}$.

# 4 EXPERIMENT

## 4.1 EXPERIMENT SETUP

**Backbone LLM.** Following prior works (Wang et al., 2024; Muennighoff et al., 2024), we adopt **Mistral-7B** (Jiang et al., 2023) as the primary backbone and further validate it on **Qwen2.5-7B** (Yang et al., 2024).

**Training Details.** For training data, we use the dataset from (Li et al., 2025), which integrates supervised pairs and hard negatives for contrastive learning across diverse tasks. Due to computational limits, we sample 20% (0.2M) data for training. Following (Wang et al., 2024; Li et al., 2025), we fine-tune the LLM as an embedding model with LoRA and contrastive loss, applying task-specific instruction templates. Specifically, for a given query $q$, we format it as $q^{+} = $ Instruct: {task_definition}$\backslash n$Query: {$q$}. Detailed hyperparameters and instructions are in Appendices C and J.

**Evaluation.** We evaluate on MTEB (English, v2) (Enevoldsen et al., 2025), covering 41 datasets across 7 task types, reporting official leaderboard scores when available. Following InBedder (Peng et al., 2024), we also evaluate on INTENTEMOTION and NYTCLUSTERING to test the instruction-following ability of embedding models. For more extensive comparison, we have also evaluated on TREC datasets used in FollowIR (Weller et al., 2025) and BEIR (Thakur et al., 2021), detailed results can be found in Appendix H.

**Comparison Methods.** We compare GIRCSE against four categories of text embedding models. (1) **Non-LLM methods** including encoder-based models such as **E5-Large** (Wang et al., 2022), **GTE-Large** (Li et al., 2023b), **BGE-Large** (Li et al., 2023a), and **UAE-Large** (Li & Li, 2024). (2) **LLM-based methods** are instruction-tuned LLM embeddings, including **LLM2Vec** (BehnamGhader et al., 2024), **GritLM** (Muennighoff et al., 2024), **E5-Mistral** (Wang et al., 2024), **NV-Embed-v1** (Lee et al., 2024), **SFR-Embedding-2** (Meng* et al., 2024), and **gte-Qwen2** (Li et al., 2023b). (3) **Generative embeddings** cover (i) two-stage approaches that expand text with an auxiliary LLM before re-encoding (see Appendix F for detail) and (ii) the end-to-end generative model **Inbedder** (Peng et al., 2024). (4) **Fair Baselines** are included by re-implementing two paradigms on the same training data for fair comparison: (i) **Causal-EOS** (causal attention + EOS pooling) and (ii) **Bidirectional-Avg** (bidirectional attention + average pooling), equivalent to **E5-Mistral** and **GritLM** respectively but trained with less data.

## 4.2 MAIN RESULTS

Table 2 reports the performance comparison across MTEB tasks and instruction-following benchmarks. We highlight the following observations:

**Trade-off between generic tasks and instruction following.** State-of-the-art non-generative embedding models achieve strong results on generic MTEB tasks but lag behind on instruction-following benchmarks. For example, **gte-QWEN2** performs competitively on MTEB (rank 1) but drops notably on instruction-following tasks (rank 18). Similarly, **E5-Mistral** ranks 4 on MTEB but falls to 10 on instruction following. In contrast, generative embedding approaches such as **Inbedder** reverse this trend, achieving top instruction-following performance (rank 1), since it is explicitly trained for this setting, but performing poorly on MTEB (rank 20). A comparable trade-off is also observed in two-stage generative variants of non-generative models. For instance, **E5-Mistral (w/ gen)** improves on instruction following (rank 10 → 5) but degrades on MTEB (rank 4 → 12). Similar patterns are also observed for **E5-Large (w/ gen)** and **GritLM (w/ gen)**.

**GIRCSE overcomes trade-off and strikes a balanced performance.** Unlike prior methods, GIRCSE delivers consistently strong results across both task categories. It not only outperforms fair baselines and competitive embedding models (e.g., GritLM, LLM2Vec), but also avoids the severe trade-offs observed in existing approaches. Specifically, GIRCSE ranks within the **top 5–6 on MTEB** and **top 2–3 on instruction-following tasks**, leading to the **best overall rankings of 3.5 and 4.5** across benchmarks. Remarkably, while prior SOTA methods rely on multi-million–scale

training datasets, GIRCSE achieves comparable or better performance with only **0.2M training examples**. These results highlight GIRCSE as an efficient embedding model that achieves both strong general-purpose performance and robust instruction-following ability.

Table 2: Performance on MTEB and instruction-following tasks. [†] Results obtained from the official MTEB leaderboard. **Causal-EOS**: causal attention with EOS pooling; **Bidirectional-Avg**: bidirectional attention with average pooling. Highlighted rows are our reproductions, trained on a smaller dataset (0.2M) for fair comparison. **Bold** = better than fair baselines with same backbone; * = statistically significant ($p < 0.05$). For detailed performance of each MTEB dataset, please refer to Appendix I.

| Task | Size | Vol. | Backbone | MTEB (English, v2) | | | | | | | | Instruct Following | | | Overall Rank |
| | | | | Retr. | Rerank. | Clust. | PairClass. | Class. | STS | Summ. | Avg. (Rank) | IntEmo | NYT | Avg. (Rank) | |
| # of datasets → | - | - | - | 10 | 2 | 8 | 3 | 8 | 9 | 1 | 41 | 1 | 1 | 2 | |
| *LLM-based: Causal-EOS* | | | | | | | | | | | | | | | |
| E5-Mistral[†] | 7B | 1.8M | Mistral | 57.62 | 49.78 | 51.44 | 88.42 | 79.85 | 84.32 | 36.57 | 67.97 (4) | 48.84 | 65.06 | 56.95 (10) | 7.0 |
| SFR-Embedding-2[†] | 7B | 1.7M | Mistral | 53.75 | 48.99 | 59.39 | 88.09 | 90.54 | 80.86 | 35.54 | 69.82 (2) | 50.49 | 60.54 | 55.52 (11) | 6.5 |
| gte-Qwen2[†] | 7B | 800M | QWEN2 | 58.09 | 50.47 | 58.97 | 85.90 | 88.52 | 82.69 | 35.74 | 70.72 (1) | 52.62 | 17.52 | 35.07 (18) | 9.5 |
| Fair Baseline | 7B | 0.2M | Mistral | 55.24 | 49.21 | 54.28 | 85.65 | 84.36 | 73.98 | 36.31 | 66.32 (10) | 35.33 | 58.76 | 47.05 (15) | 12.5 |
| Fair Baseline | 7B | 0.2M | QWEN2 | 51.10 | 47.49 | 55.26 | 84.46 | 80.10 | 74.71 | 33.21 | 64.18 (17) | 66.14 | 14.71 | 40.42 (17) | 17.0 |
| *LLM-based: Bidirectional-Avg* | | | | | | | | | | | | | | | |
| LLM2Vec[†] | 7B | 1.5M | Mistral | 51.27 | 47.74 | 44.10 | 87.99 | 79.74 | 83.70 | 31.05 | 64.57 (15) | 51.66 | 61.38 | 56.52 (8) | 11.5 |
| GritLM[†] | 7B | 2M | Mistral | 54.95 | 49.59 | 50.82 | 87.29 | 81.25 | 83.03 | 35.65 | 67.07 (7) | 39.30 | 79.25 | 59.28 (6) | 6.5 |
| NV-Embed-v1[†] | 7B | 1.1M | Mistral | 60.13 | 49.16 | 49.50 | 87.05 | 84.11 | 82.20 | 31.40 | 68.32 (3) | 52.61 | 60.62 | 56.62 (7) | 5.0 |
| Fair Baseline | 7B | 0.2M | Mistral | 55.41 | 48.74 | 54.57 | 86.34 | 84.94 | 75.87 | 36.09 | 66.96 (8) | 21.45 | 66.42 | 43.94 (16) | 12.0 |
| Fair Baseline | 7B | 0.2M | QWEN2 | 52.99 | 47.11 | 54.75 | 83.31 | 82.66 | 72.81 | 35.30 | 64.97 (16) | 43.26 | 65.21 | 54.24 (13) | 14.5 |
| *LLM-based: Two-Stage Generative Embedding* | | | | | | | | | | | | | | | |
| E5-Large (w/ gen.) | 0.3B | 1B | BERT | 45.06 | 43.87 | 45.37 | 81.02 | 72.70 | 77.35 | 31.59 | 59.85 (19) | 51.34 | 51.67 | 51.51 (12) | 15.5 |
| E5-Mistral (w/ gen.) | 7B | 1.8M | Mistral | 57.20 | 49.18 | 53.02 | 84.26 | 75.97 | 79.52 | 31.73 | 65.92 (12) | 58.64 | 60.89 | 59.77 (5) | 8.5 |
| GritLM (w/ gen.) | 7B | 2M | Mistral | 56.48 | 49.45 | 52.03 | 83.36 | 77.77 | 79.66 | 32.82 | 65.90 (11) | 51.16 | 70.50 | 60.83 (4) | 7.5 |
| *LLM-based: End2End Generative Embedding* | | | | | | | | | | | | | | | |
| Inbedder | 7B | 0.2M | LLaMA2 | 12.50 | 39.21 | 51.24 | 61.17 | 72.41 | 74.41 | 17.24 | 50.32 (20) | 89.68 | 64.65 | 77.17 (1) | 10.5 |
| GIRCSE | 7B | 0.2M | Mistral | **57.10** | 48.88 | **56.26** | 86.18 | **85.33** | 76.37 | 33.56 | **67.83\*** (5) | 52.19 | 73.75 | 62.97 (2) | 3.5 |
| GIRCSE | 7B | 0.2M | QWEN2 | **55.16** | **49.28** | **56.62** | **85.17** | **86.69** | **76.30** | **35.42** | **67.67\*** (6) | 64.92 | 60.04 | 62.48 (3) | 4.5 |

## 4.3 ABLATION STUDY

To better understand the contributions of different components in GIRCSE, Table 3 represents an ablation study on generative embedding, stepwise loss (SL), and iterative refinement (IR). Starting from the variant without generation (i.e., the *Causal-EOS* baseline), we observe a substantial drop in performance across both MTEB and instruction-following tasks. Incorporating generative embedding alone yields consistent improvements across nearly all tasks. Adding SL provides further gains, particularly for classification and summarization, while the combination of SL and IR achieves the strongest overall performance. Overall, these results validate the effectiveness of our design in GIRCSE.

Table 3: Ablation study of GIRCSE with generative embedding (Gen.), stepwise loss (SL), and iterative refinement (IR). The variant without generation corresponds to the Causal-EOS baseline. Results are reported using the Mistral-7B backbone trained on 50K samples.

| Gen. | SL | IR | MTEB (English, v2) | | | | | | | | Instruct Following | | |
| | | | Retr. | Rerank. | Clust. | PairCls. | Class. | STS | Summ. | Avg. | IntEmo | NYT | Avg. |
| ✗ | ✗ | ✗ | 50.55 | 48.97 | 49.91 | 85.27 | 80.36 | 75.76 | 34.02 | 63.84 | 33.11 | 58.76 | 47.05 |
| ✓ | ✗ | ✗ | 53.17 | 48.32 | 52.74 | 84.75 | 78.34 | 78.70 | 33.86 | 65.21 | 48.00 | 64.93 | 56.47 |
| ✓ | ✓ | ✗ | 54.97 | 48.86 | 52.07 | 85.04 | 78.63 | 78.87 | 35.28 | 65.69 | 53.88 | 66.37 | 60.13 |
| ✓ | ✓ | ✓ | 55.53 | 48.26 | 53.71 | 84.87 | 79.53 | 78.93 | 34.19 | **66.27** | 62.70 | 73.75 | **62.97** |

## 5 ANALYSIS ON GENERATED TOKENS

While Section 4.3 highlights the importance of generative embedding, it remains unclear how the generation process itself translates to the improved performance. To address this, we present a thorough analysis of the generation process. In Section 5.1, we first discuss how the embedding quality changes by varying the number of generated tokens at inference. Next, in Section 5.2, we conduct a qualitative analysis to understand what tokens are generated and how they evolve under different instructions. This analysis clarifies how iterative generation improves performance and what semantic signals are encoded in the embedding space.

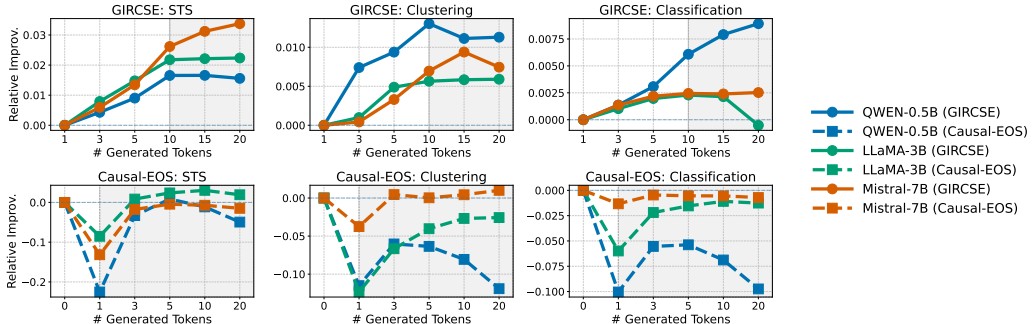

Figure 2: Effect of generation length at inference. **Top**: GIRCSE consistently improves with longer generations (10–20 tokens) despite been trained on only 5 tokens. **Bottom**: Baseline models show degraded or fluctuated performance across generation lengths. Gray area indicates configurations beyond training length.

## 5.1 EFFECT OF GENERATION LENGTH AT INFERENCE

We first examine how performance varies with the number of generated tokens $K$ at inference. We evaluate $K \in \{1, 3, 5, 10, 15, 20\}$ and compare against the non-generative baseline *Causal-EOS*. Results (Fig. 2) are reported on three LLM backbones and three representative MTEB tasks, with relative improvements measured against $K = 1$ for GIRCSE and against the no-generation baseline for *Causal-EOS*. For a more comprehensive analysis, we further evaluate GIRCSE trained with two additional backbones: **QWEN2.5-0.5B** (Yang et al., 2024) and **LLaMA3.2-3B** (Dubey et al., 2024). We have the following two key findings:

**(1) GIRCSE exhibits test-time scaling for embeddings.** Increasing $K$ consistently improves performance across diverse tasks (e.g., STS, clustering, classification) and across model sizes. In contrast, the non-generative method (*Causal-EOS*) does not benefit from additional generation and often degrades in performance. This suggests that GIRCSE successfully learns an iterative refinement mechanism that converts additional inference computation into stronger semantic representations—analogous to test-time compute scaling in reasoning LLMs (Muennighoff et al., 2025), but novel in the context of embedding models.

**(2) ICR enables GIRCSE to generalize beyond training configurations.** Although GIRCSE is trained with $K = 5$, its performance improves monotonically within the training regime ($K = 1, 3, 5$) and continues to improve even beyond it ($K = 10, 15, 20$). This extrapolation capability suggests that our ICR training objective enables the learned refinement process to generalize beyond the training configuration, allowing GIRCSE to continue improving with additional inference steps. Overall, GIRCSE establishes test-time scaling as a new paradigm for embedding models, enabling controllable and training-free performance gains through adjustable generation length.

## 5.2 QUALITATIVE ANALYSIS ON GENERATED TOKENS

Having shown that generating more tokens improves performance, we next ask: what do these tokens capture? We analyze generations for the sentence "Why is it so hard to track down this card?" under two prompts: representing intention and emotion. At each generation step $k$, we collect the top-30 candidates from the soft token distribution $\mathbf{s}_k$, aggregate across steps, and report most frequent tokens in Table 4, alongside results from GIRCSE before contrastive fine-tuning. Before fine-tuning, GIRCSE (before FT) often yields generic or semantically weak tokens. After fine-tuning, we observe progressive semantic refinement that aligns with the results in Section 5.1. At early steps (1–5), GIRCSE generates core content words (e.g., why, hard, card). While at later steps, the outputs diverge by different instructions: intention produces tokens such as seek, elusive, inquiry, and emotion yields tokens like frustrating, struggle. This suggests multi-step generation acts as a semantic chain of thought, iteratively steering representations toward nuanced, instruction-aligned regions of the embedding space.

Table 4: Qualitative analysis of generated tokens. Gray indicates generic/stopword-like tokens. Yellow marks core input-related tokens shared across instructions. Instruction-specific expansions are shown in Green (*intention*) and Red (*emotion*).

| Input Sentence | *"Why is it so hard to track down this card?"* | |
|---|---|---|
| **Instruction** | *"Represent the **intention** of this text."* | *"Represent the **emotion** of this text."* |
| **GIRCSE (Before FT)** | this, so, do, how, i, is, it, the, we, what | why, how, what, this, it, is, you, can |
| **GIRCSE (Step 1–5)** | why, is, it, hard, track/tracking, card, this | why, is, it, hard, track, tracking, card, this, so, difficult |
| **GIRCSE (Step 6–10)** | [prev.] + seek, elusive, so | [prev.] + frustrating, tough, persistent, struggle, challenging |
| **GIRCSE (Step 11–20)** | [prev.] + question, inquiry | [prev.] + perseverance, stuck, complicated |

# 6 DISCUSSION OF ROBUSTNESS AND LEARNING EFFICIENCY

To assess the robustness and learning efficiency of our method, we conduct comprehensive experiments across varying data scales and backbone architectures. Specifically, we train different models with {50K, 100K, 200K} training samples using three widely adopted open-source LLMs as base models: Qwen-0.5B, Llama-3B, and Mistral-7B. We compare GIRCSE against two fair baselines: *Causal-EOS* and *Bidirectional-Avg*. Fig. 3 shows that our method consistently outperforms both baselines across all data scales and model sizes. In particular, when trained with only 50K samples, our method improves over *Causal-EOS* by +5.7 points on Qwen-0.5B (61.2% vs. 55.5%) and by +2.8 points on Llama-3B (65.5% vs. 62.7%). Even with stronger backbones such as Mistral-7B, our approach still yields gains of +2.4 points (66.2% vs. 63.8%). The performance gap becomes more pronounced when the training data is limited. These findings indicate that our approach not only achieves superior performance across different scales of model size but also learns more effectively under limited training data.

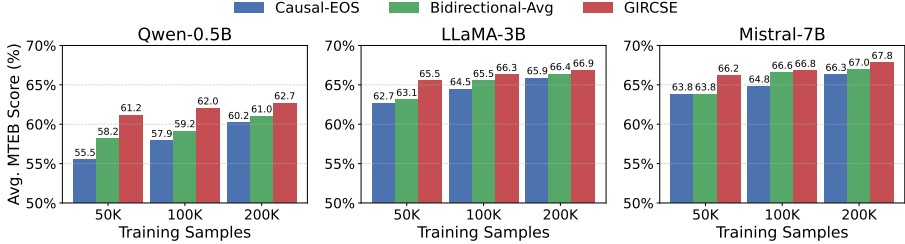

Figure 3: Comparison of average MTEB scores (%) between GIRCSE and two fair baselines across three backbone LLMs and varying training sample sizes. GIRCSE consistently delivers superior performance, especially under limited-data settings.

# 7 CONCLUSION

We presented GIRCSE, a generative embedding framework that leverages autoregressive refinement to move beyond single-pass LLM encoders. By generating soft refinement tokens and training with iterative contrastive refinement, GIRCSE enables embeddings to progressively distill semantics rather than compressing them in one step. Experiments show that GIRCSE achieves state-of-the-art or competitive performance across benchmarks while introducing a novel scaling property: embedding quality improves with additional refinement steps at test time. These results highlight autoregressive generation as a powerful mechanism for embedding optimization and open new directions for scalable, semantically rich representations.

ACKNOWLEDGMENTS

This material is based upon work supported by National Science and Technology Council, ROC under grant number 114-2221-E-002-134-MY3 and by Taiwan Centers of Excellence (TCE)

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

## A   USE-OF-LLMS

In this work, we utilized large language models (LLMs) as part of the core research methodology. Specifically, we fine-tuned existing open-source LLMs (e.g., LLaMA-3 and Mistral) to develop embedding models. These pre-trained models served as the foundation for our experiments, and our main contributions build upon their architectures and representations. Additionally, an LLM-based assistant (OpenAI GPT-5) was used for minor writing support, including grammar checking and improving manuscript readability. All decisions regarding research design, fine-tuning strategies, experimental setup, and final interpretations were made solely by the authors.

## B   REPRODUCIBILITY FOR EMBEDDING MODELS

To facilitate reproducibility of our experiments, we provide links to all open-sourced embedding models used in this paper in Table 5. These links allow researchers to directly access the exact model checkpoints we relied on.

Table 5: List of models with links for reproducibility.

| Model | Link |
|---|---|
| E5-Large | huggingface.co/intfloat/e5-large |
| GTE-Large | huggingface.co/thenlper/gte-large |
| BGE-Large | huggingface.co/BAAI/bge-large-en |
| UAE-Large | huggingface.co/WhereIsAI/UAE-Large-V1 |
| E5-Mistral | huggingface.co/intfloat/e5-mistral-7b-instruct |
| SFR-Embedding-2 | huggingface.co/Salesforce/SFR-Embedding-2_R |
| gte-Qwen2 | huggingface.co/Alibaba-NLP/gte-Qwen2-7B-instruct |
| LLM2Vec | github.com/McGill-NLP/LLM2Vec |
| GritLM | huggingface.co/GritLM/GritLM-7B |
| NV-Embed-v1 | huggingface.co/nvidia/NV-Embed-v1 |

## C   HYPERPARAMETER SETTINGS

For both GIRCSE and the baseline models we re-implement for comparison, we use nearly identical fine-tuning hyperparameters across different model sizes (QWEN-0.5B, Llama-3B, and Mistral-7B) and training data scales (50k, 100k, and 200k examples).

We adopt Low-Rank Adaptation (LoRA) (Hu et al., 2022) for efficient fine-tuning, setting the rank to 64 and the scaling factor $\alpha$ to 32. The default learning rate is 1e-5 with a warmup ratio of 0.1. The only exception is Llama-3B, for which we use a learning rate of 1e-4 to address convergence issues. For other hyperparameters, we set the temperature of the contrastive loss (Eq. (7)) to 0.02 across all models, and the weighting coefficient $\lambda$ in GIRCSE for balancing contrastive alignment and refinement regularization to 1. Due to limited computational resources, we train with a batch size of 2 and accumulate gradients over 8 steps, resulting in an effective batch size of 16. All models are fine-tuned for a single epoch.

For GIRCSE, we set the number of generated tokens $K$ to 5 during training to avoid the high computational cost of multiple autoregressive forward passes. During inference, we increase $K$ to 20 to enable longer generations and improved embedding refinement, while mitigating the computational overhead using KV-cache techniques.

## D   SCALABILITY ANALYSIS

While GIRCSE offers superior representation quality, a natural concern arises regarding its computational efficiency. We acknowledge that GIRCSE introduces additional overhead compared to

traditional embedding models due to its generative process; a detailed analysis of theoretical computation and memory costs relative to the discriminative embedding paradigm is provided in Appendix E. Nevertheless, this overhead can be substantially mitigated through the use of KV caching techniques (Devoto et al., 2024). As shown in Table 6, GIRCSE without caching requires 2.0–6.0× more FLOPs across different sequence lengths due to auto-regressive computation. In contrast, with caching enabled, the FLOPs are effectively reduced to baseline levels ($\approx 1.0\times$), while memory consumption remains comparable to traditional methods. These results demonstrate that caching not only ensures scalability but also makes our approach practical for real-world deployment.

Table 6: Computational efficiency comparison across sequence lengths (512, 1024, 2048) and generation budgets $k$. Lower is better ($\downarrow$) for both FLOPs and memory. GIRCSE without caching (red) incurs significant computational overhead due to auto-regressive processing, whereas KV caching (blue) dramatically mitigates this cost. Multipliers in parentheses show overhead relative to Causal-EOS method.

| Method | k | FLOPs (T) ↓ | | | Memory (GB) ↓ | | |
|---|---|---|---|---|---|---|---|
| | | 512 | 1K | 2K | 512 | 1K | 2K |
| **Causal-EOS** | – | 7.33 | 14.65 | 29.30 | 13.89 | 14.34 | 15.11 |
| **Bidirectional-Avg** | – | 7.33 | 14.65 | 29.30 | 14.03 | 14.45 | 15.24 |
| **GIRCSE (w/o cache)** | 1 | 14.67 (2.00×) | 29.32 (2.00×) | 58.62 (2.00×) | 13.72 | 13.90 | 14.28 |
| | 3 | 29.39 (4.01×) | 58.70 (4.01×) | 117.31 (4.00×) | 13.74 | 13.92 | 14.29 |
| | 5 | 44.17 (6.02×) | 88.13 (6.02×) | 176.04 (6.01×) | 13.75 | 13.94 | 14.30 |
| **GIRCSE (w/ cache)** | 1 | **7.34** (1.00×) | **14.67** (1.00×) | **29.32** (1.00×) | 13.73 | 13.91 | 14.32 |
| | 3 | **7.37** (1.01×) | **14.70** (1.00×) | **29.35** (1.00×) | 13.75 | 13.93 | 14.34 |
| | 5 | **7.40** (1.01×) | **14.73** (1.01×) | **29.38** (1.00×) | 13.77 | 13.96 | 14.35 |

# E  THEORETICAL COMPUTATIONAL AND MEMORY COST ANALYSIS

To better analyze the additional training and inference cost introduced by GIRCSE, we compare the computational and memory complexity of the proposed generative embedding framework against the conventional discriminative embedding paradigm. In the baseline discriminative case, the encoder processes an input of length $N$, leading to a per-layer cost dominated by self-attention of order $O(N^2 d)$ and memory footprint $O(LN^2)$, where $d$ is the embedding dimension and $L$ is the number of layers.

In the generative framework, $K$ auxiliary soft tokens are generated autoregressively. Each generation step requires an encoder forward pass over $N + j$ tokens ($j = 0, \ldots, K - 1$) followed by a vocabulary softmax of cost $O(d|\mathcal{V}|)$. After generation, a final encoder pass is performed over the extended sequence of length $N + K$. The total attention-dominated computation ratio with respect to the baseline is:

$$R_{\text{computation}} = \frac{C_{\text{gen}}}{C_{\text{base}}} = \frac{(K+1)N^2 + NK(K+1) + \frac{K(K+1)(2K+1)}{6}}{N^2}, \tag{9}$$

which simplifies to $R_{\text{computation}} \approx K+1$ when $K \ll N$. The additional softmax operations contribute $K\,O(d|\mathcal{V}|)$, which is typically small compared to the quadratic encoder cost unless $N$ is short or $|\mathcal{V}|$ is very large.

In terms of memory, peak training-time activation usage is dominated by the final encoder pass over $N + K$ tokens. Thus, the relative peak memory ratio is:

$$R_{\text{memory}} = \frac{M_{\text{gen}}}{M_{\text{base}}} \approx \frac{(N+K)^2}{N^2}. \tag{10}$$

This indicates that while the generative embedding framework incurs roughly $K$ additional encoder passes in computation, the increase in peak memory is modest, scaling quadratically with the extended sequence length $N + K$.

## F    DETAIL IMPLEMENTATION OF TWO-STAGE GENERATION EMBEDDING

The two-stage generation embedding approach enhances representation quality by introducing an intermediate expansion step before re-encoding.

In the first stage, an auxiliary large language model (LLM) is prompted to generate a short augmentation of the input, detailed expansion prompt can be found in Table 7. The prompt instructs the model to output only the augmentation, within a fixed token budget, without explanations or additional formatting. This augmentation is designed to highlight or enrich semantic information that may be useful for downstream tasks.

In the second stage, the original instruction and text is concatenated with the generated augmentation, and the combined sequence is re-encoded into an embedding. This two-step process allows the encoder to capture a more informative and contextually aligned representation than directly embedding the raw text alone.

Table 7: LLM expansion prompt used for two-stage generation methods.

| **Input:** |
| --- |
| Given the INSTRUCTION and the TEXT, produce a helpful augmentation that, when concatenated to the original TEXT and embedded, is likely to improve embedding quality for the instruction's task. |
| Do not explain yourself or output anything other than the augmentation. |
| Your answer must be written within {256} tokens. |
| INSTRUCTION: {instruction} |
| TEXT: {text} |

## G    TRAINING STABILITY ANALYSIS

A potential concern when optimizing models that involve the generation of soft tokens is the risk of gradient instability. To empirically validate the stability of our proposed model, GIRCSE, we monitor its training dynamics. We present the training loss curve in Figure 4 and the L2 norm of the gradients in Figure 5.

As shown in the figures, the training loss (Figure 4) demonstrates a smooth and consistent decrease, indicating stable convergence. Furthermore, the gradient norm (Figure 5) converges normally throughout the training process.

## H    MORE EVALUATION ON NANOBEIR AND TREC BENCHMARKS

In this section, we provide additional evaluation results on the NanoBEIR and TREC benchmarks. These benchmarks complement the main paper by covering a wider range of retrieval tasks and domains. The results, presented in Table 8 and Table 9, highlight consistent trends with our main findings, further demonstrating the robustness and generalization ability of the proposed model compared to strong baselines.

## I    FULL MTEB PERFORMANCE

In the main paper, we reported averaged performance across task categories for clarity. Here, we provide the full per-dataset results on the MTEB benchmark, covering all tasks included in our evaluation. The detailed scores in Table 10 allow a more granular comparison across individual datasets and complement the averaged results presented in the main section.

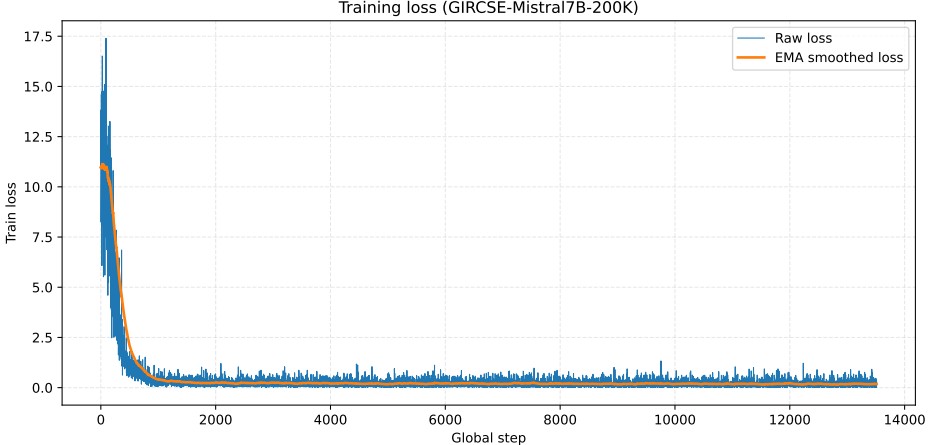

Figure 4: Training loss curve of GIRCSE.

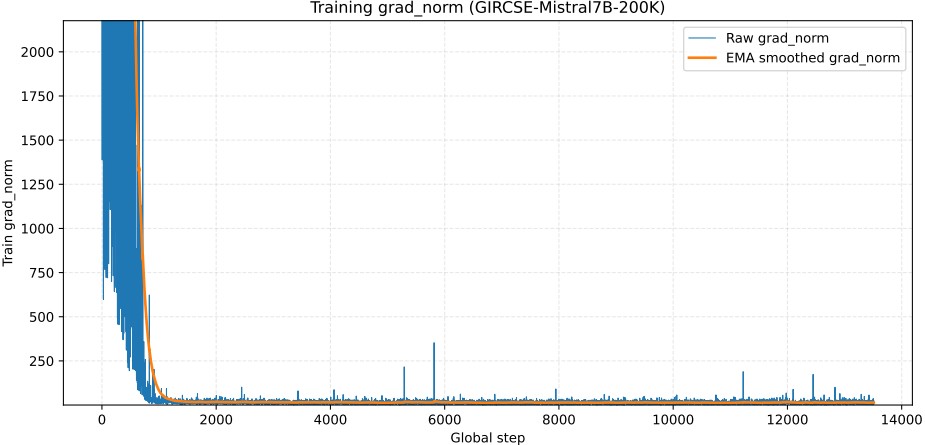

Figure 5: Training gradient norm (L2) of GIRCSE, plotted with the top 2% of outliers removed for clarity.

Table 8: Performance comparison on NanoBEIR benchmark.

| Dataset | E5-Mistral | SFR-Embedding-2 | gte-Qwen2 | LLM2Vec | GritLM | NV-Embed-v1 | GIRCSE | GIRCSE |
|---|---|---|---|---|---|---|---|---|
| Size | 7B | 7B | 7B | 7B | 7B | 7B | 7B | 7B |
| Vol. | 1.8M | 1.7M | 800M | 1.5M | 2M | 1.1M | 0.2M | 0.2M |
| Backbone | Mistral | Mistral | QWEN2 | Mistral | Mistral | Mistral | Mistral | QWEN2 |
| ArguAna | 60.13 | 64.33 | 72.37 | 54.41 | 67.72 | 68.00 | 70.18 | 70.88 |
| ClimateFever | 41.75 | 44.12 | 33.83 | 38.09 | 37.45 | 42.93 | 36.83 | 31.66 |
| DBPedia | 71.59 | 73.81 | 72.14 | 70.78 | 68.15 | 71.44 | 71.01 | 69.81 |
| FEVER | 95.30 | 95.18 | 80.71 | 96.94 | 94.38 | 96.15 | 94.23 | 95.41 |
| FiQA | 60.50 | 65.08 | 68.23 | 61.46 | 65.98 | 68.07 | 60.79 | 55.90 |
| HotpotQA | 86.16 | 91.64 | 82.06 | 88.14 | 91.09 | 91.05 | 88.10 | 88.31 |
| MSMARCO | 66.58 | 66.51 | 70.67 | 64.30 | 62.82 | 67.80 | 64.03 | 61.85 |
| NFCorpus | 32.36 | 38.99 | 37.67 | 39.19 | 40.39 | 38.54 | 40.75 | 36.57 |
| NQ | 76.64 | 82.20 | 80.99 | 83.81 | 82.91 | 87.11 | 78.57 | 70.01 |
| Quora | 96.28 | 95.48 | 97.44 | 95.64 | 95.76 | 91.59 | 92.73 | 88.76 |
| SCIDOCS | 36.00 | 48.18 | 50.76 | 43.72 | 46.32 | 38.11 | 44.27 | 43.97 |
| SciFact | 78.30 | 89.67 | 67.58 | 79.41 | 81.18 | 78.95 | 80.90 | 81.70 |
| Touche2020 | 51.72 | 56.66 | 39.44 | 49.46 | 51.89 | 58.51 | 46.09 | 39.03 |
| Avg. | 65.64 | 70.14 | 65.68 | 66.57 | 68.16 | 69.10 | 66.81 | 64.14 |

Table 9: Performance comparison on TREC datasets. [†] Results obtained from (Weller et al., 2025) Best results per task (i.e., column) are in **bold**, with second best results are in underline.

| Model | Robust04 MAP | News21 nDCG | Core17 MAP | Avg. |
|---|---|---|---|---|
| ***No-Instruction IR*** | | | | |
| E5-base-v2[†] | 13.4 | 20.9 | 14.0 | 16.1 |
| Contriever[†] | 19.7 | 22.9 | 15.3 | 19.3 |
| MonoBERT[†] | 21.0 | 25.1 | 18.4 | 21.5 |
| BM25[†] | 12.1 | 19.3 | 8.1 | 13.2 |
| MonoT5-base[†] | 15.7 | 11.0 | 12.2 | 13.0 |
| E5-large-v2[†] | 17.4 | 24.3 | 17.0 | 19.6 |
| MonoT5-3B[†] | 27.3 | 16.5 | 18.2 | 20.7 |
| ***Instruction-IR*** | | | | |
| TART-Contriever[†] | 14.3 | 21.8 | 13.3 | 16.5 |
| INSTRUCTOR-base[†] | 17.2 | 22.1 | 15.5 | 18.3 |
| E5-mistral[†] | 23.1 | 27.8 | 18.3 | 23.1 |
| BGE-base[†] | 16.8 | 20.0 | 14.6 | 17.1 |
| INSTRUCTOR-xl[†] | 19.7 | 26.1 | 16.8 | 20.9 |
| BGE-large[†] | 17.5 | 22.3 | 15.0 | 18.3 |
| GritLM-7B[†] | **28.6** | 24.4 | 20.8 | 24.6 |
| TART-FLAN-T5-xl[†] | 24.6 | 12.8 | 17.0 | 18.1 |
| ***APIs*** | | | | |
| OpenAI v3 Large[†] | 27.2 | 27.2 | 21.6 | 25.3 |
| Cohere v3 English[†] | 22.3 | 28.3 | 20.6 | 23.7 |
| Google Gecko[†] | 23.3 | 29.5 | **23.2** | 25.3 |
| ***Instruct LMs*** | | | | |
| FLAN-T5-base[†] | 6.4 | 6.1 | 6.5 | 6.3 |
| Llama-2-7B-chat[†] | 6.3 | 1.7 | 5.4 | 4.5 |
| FLAN-T5-large[†] | 14.7 | 8.0 | 11.4 | 11.4 |
| GritLM-Reranker[†] | 9.7 | 10.2 | 9.8 | 9.9 |
| Mistral-7B-instruct[†] | 23.2 | 27.2 | 19.7 | 23.4 |
| FollowIR-7B[†] | 24.8 | **29.6** | 20.0 | 24.8 |
| ***End2End Generative Embedding*** | | | | |
| GIRCSE-Mistral-7B | 27.9 | 26.8 | 23.0 | **25.9** |
| GIRCSE-Qwen2-7B | 23.9 | 24.1 | 21.0 | 23.0 |

## J    FULL DATASET INSTRUCTIONS

This section provides the complete set of natural language instructions used with the datasets in our experiments. We include both the instructions for the MTEB benchmark datasets and those for the Instruction Following tasks. These instructions define the intended task for each dataset and serve as the input prompts during embedding evaluation. The full text of the instructions is listed in Table 11 and Table 12, respectively.

Table 10: Full evaluation results across MTEB tasks for GIRCSE, Causal-EOS, and Bidirectional-Avg with Mistral and Qwen backbones. Best results per task (i.e., row) are in **bold**.

| Task | GIRCSE | | Causal-EOS | | Bidirectional-Avg | |
|---|---|---|---|---|---|---|
| | **Mistral** | **Qwen** | **Mistral** | **Qwen** | **Mistral** | **Qwen** |
| MindSmallReranking | 0.294 | **0.321** | 0.307 | 0.308 | 0.302 | 0.311 |
| AskUbuntuDupQuestions | **0.684** | 0.664 | 0.677 | 0.642 | 0.673 | 0.631 |
| TwitterSemEval2015 | 0.769 | 0.747 | 0.760 | 0.732 | **0.786** | 0.707 |
| StackExchangeClusteringP2P.v2 | 0.499 | **0.508** | 0.457 | 0.463 | 0.439 | 0.479 |
| BiorxivClusteringP2P.v2 | 0.487 | **0.497** | 0.439 | 0.431 | 0.459 | 0.472 |
| SICK-R | 0.727 | 0.742 | 0.731 | 0.741 | **0.753** | 0.746 |
| ToxicConversationsClassification | 0.841 | **0.870** | 0.781 | 0.735 | 0.811 | 0.760 |
| TweetSentimentExtractionClassification | 0.762 | **0.775** | 0.730 | 0.703 | 0.744 | 0.723 |
| TwentyNewsgroupsClustering.v2 | 0.611 | **0.627** | 0.598 | 0.582 | 0.624 | 0.569 |
| STS15 | **0.845** | 0.833 | 0.835 | 0.817 | 0.839 | 0.815 |
| MTOPDomainClassification | 0.957 | **0.967** | 0.958 | 0.960 | 0.963 | 0.944 |
| STSBenchmark | **0.832** | 0.820 | 0.807 | 0.814 | 0.823 | 0.773 |
| STS17 | 0.801 | 0.795 | 0.774 | **0.821** | 0.805 | 0.806 |
| ClimateFEVERHardNegatives | 0.268 | 0.261 | 0.220 | 0.222 | 0.227 | **0.269** |
| HotpotQAHardNegatives | 0.729 | **0.737** | 0.738 | 0.708 | 0.735 | 0.686 |
| FiQA2018 | **0.552** | 0.489 | 0.486 | 0.480 | 0.521 | 0.440 |
| CQADupstackGamingRetrieval | 0.641 | **0.656** | 0.587 | 0.576 | 0.615 | 0.574 |
| SprintDuplicateQuestions | 0.943 | **0.949** | 0.946 | 0.940 | 0.939 | 0.942 |
| ArguAna | 0.699 | 0.692 | 0.694 | 0.574 | **0.703** | 0.662 |
| MassiveIntentClassification | 0.772 | **0.781** | 0.775 | 0.757 | 0.772 | 0.756 |
| SCIDOCS | 0.237 | **0.250** | 0.234 | 0.231 | 0.242 | 0.231 |
| STS22.v2 | **0.690** | 0.665 | 0.607 | 0.447 | 0.645 | 0.679 |
| STS12 | 0.730 | **0.740** | 0.724 | 0.731 | 0.738 | 0.708 |
| STS13 | 0.717 | 0.725 | 0.704 | **0.762** | 0.714 | 0.581 |
| MedrxivClusteringP2P.v2 | 0.429 | **0.433** | 0.386 | 0.338 | 0.387 | 0.418 |
| MassiveScenarioClassification | 0.793 | **0.802** | 0.807 | 0.778 | 0.798 | 0.790 |
| STS14 | **0.746** | 0.744 | 0.710 | 0.739 | 0.741 | 0.665 |
| ArXivHierarchicalClusteringP2P | 0.645 | 0.630 | 0.639 | **0.646** | 0.634 | 0.639 |
| ImdbClassification | 0.960 | **0.962** | 0.955 | 0.906 | 0.959 | 0.949 |
| Banking77Classification | 0.855 | **0.861** | 0.849 | 0.838 | 0.853 | 0.823 |
| Touche2020Retrieval.v3 | 0.478 | 0.391 | **0.490** | 0.459 | 0.452 | 0.470 |
| SummEvalSummarization.v2 | 0.336 | 0.354 | **0.363** | 0.332 | 0.361 | 0.353 |
| TwitterURLCorpus | **0.873** | 0.859 | 0.864 | 0.862 | 0.865 | 0.850 |
| AmazonCounterfactualClassification | 0.887 | **0.918** | 0.893 | 0.731 | 0.896 | 0.867 |
| MedrxivClusteringS2S.v2 | 0.423 | **0.428** | 0.421 | 0.409 | 0.423 | 0.403 |
| StackExchangeClustering.v2 | 0.759 | **0.772** | 0.758 | 0.747 | 0.755 | 0.742 |
| FEVERHardNegatives | 0.853 | **0.857** | 0.786 | 0.702 | 0.809 | 0.835 |
| CQADupstackUnixRetrieval | 0.522 | **0.545** | 0.493 | 0.455 | 0.504 | 0.450 |
| TRECCOVID | 0.731 | 0.640 | **0.797** | 0.702 | 0.733 | 0.682 |
| BIOSSES | 0.786 | 0.804 | 0.765 | **0.851** | 0.771 | 0.781 |
| ArXivHierarchicalClusteringS2S | 0.648 | 0.634 | 0.644 | 0.645 | 0.645 | **0.657** |

Table 11: Instructions for the corresponding datasets in the MTEB benchmark. We mainly follow the instructions from the **GritLM** paper. Note that for retrieval and reranking datasets, queries (Q) and corpus (C) documents may require different instructions, denoted as {dataset}-**Q** and {dataset}-**C**, respectively. For datasets with query instructions only (i.e., {dataset}-**Q**), no instructions are applied to the corpus.

| Dataset | Instruction |
|---|---|
| SummEvalSummarization | Given a news summary, retrieve other semantically similar summaries. |
| ArXivHierarchicalClusteringP2P | Identify the main and secondary category of Arxiv papers based on the titles and abstracts. |
| ArXivHierarchicalClusteringS2S | Identify the main and secondary category of Arxiv papers based on the titles. |
| Touche2020Retrieval.v3-**Q** | Given a question, retrieve passages that answer the question. |
| ClimateFEVERHardNegatives-**Q** | Given a claim about climate change, retrieve documents that support or refute the claim. |
| FEVERHardNegatives-**Q** | Given a claim, retrieve documents that support or refute the claim. |

**Table 11 – continued from previous page**

| Dataset | Instruction |
| --- | --- |
| HotpotQAHardNegatives-**Q** | Given a multi-hop question, retrieve documents that can help answer the question. |
| AmazonCounterfactualClassification | Classify a given Amazon customer review text as either counterfactual or not-counterfactual. |
| AmazonPolarityClassification | Classify Amazon reviews into positive or negative sentiment. |
| AmazonReviewsClassification | Classify the given Amazon review into its appropriate rating category. |
| Banking77Classification | Given a online banking query, find the corresponding intents. |
| EmotionClassification | Classify the emotion expressed in the given Twitter message into one of the six emotions: anger, fear, joy, love, sadness, and surprise. |
| ImdbClassification | Classify the sentiment expressed in the given movie review text from the IMDB dataset. |
| MassiveIntentClassification | Given a user utterance as query, find the user intents. |
| MassiveScenarioClassification | Given a user utterance as query, find the user scenarios. |
| MTOPDomainClassification | Classify the intent domain of the given utterance in task-oriented conversation. |
| MTOPIntentClassification | Classify the intent of the given utterance in task-oriented conversation. |
| ToxicConversationsClassification | Classify the given comments as either toxic or not toxic. |
| TweetSentimentExtractionClassification | Classify the sentiment of a given tweet as either positive, negative, or neutral. |
| ArxivClusteringP2P | Identify the main and secondary category of Arxiv papers based on the titles and abstracts. |
| ArxivClusteringS2S | Identify the main and secondary category of Arxiv papers based on the titles. |
| BiorxivClusteringP2P | Identify the main category of Biorxiv papers based on the titles and abstracts. |
| BiorxivClusteringS2S | Identify the main category of Biorxiv papers based on the titles. |
| MedrxivClusteringP2P | Identify the main category of Medrxiv papers based on the titles and abstracts. |
| MedrxivClusteringS2S | Identify the main category of Medrxiv papers based on the titles. |
| RedditClustering | Identify the topic or theme of Reddit posts based on the titles. |
| RedditClusteringP2P | Identify the topic or theme of Reddit posts based on the titles and posts. |
| StackExchangeClustering | Identify the topic or theme of StackExchange posts based on the titles. |
| StackExchangeClusteringP2P | Identify the topic or theme of StackExchange posts based on the given paragraphs. |
| TwentyNewsgroupsClustering | Identify the topic or theme of the given news articles. |
| SprintDuplicateQuestions | Retrieve duplicate questions from Sprint forum. |
| TwitterSemEval2015 | Retrieve tweets that are semantically similar to the given tweet. |
| TwitterURLCorpus | Retrieve tweets that are semantically similar to the given tweet. |
| AskUbuntuDupQuestions-**Q** | Retrieve duplicate questions from AskUbuntu forum. |
| AskUbuntuDupQuestions-**C** | Retrieve duplicate questions from AskUbuntu forum. |
| MindSmallReranking-**Q** | Retrieve relevant news articles based on user browsing history. |
| MindSmallReranking-**C** | Retrieve relevant news articles based on user browsing history. |
| SciDocsRR-**Q** | Given a title of a scientific paper, retrieve the titles of other relevant papers. |
| SciDocsRR-**C** | Given a title of a scientific paper, retrieve the titles of other relevant papers. |
| StackOverflowDupQuestions-**Q** | Retrieve duplicate questions from StackOverflow forum. |
| StackOverflowDupQuestions-**C** | Retrieve duplicate questions from StackOverflow forum. |
| ArguAna-**Q** | Given a claim, find documents that refute the claim. |
| ClimateFEVER-**Q** | Given a claim about climate change, retrieve documents that support or refute the claim. |
| CQADupstackRetrieval-**Q** | Given a question, retrieve detailed question descriptions from Stackexchange that are duplicates to the given question. |
| DBPedia-**Q** | Given a query, retrieve relevant entity descriptions from DBPedia. |
| FEVER-**Q** | Given a claim, retrieve documents that support or refute the claim. |
| FiQA2018-**Q** | Given a financial question, retrieve user replies that best answer the question. |

**Table 11 – continued from previous page**

| Dataset | Instruction |
|---|---|
| HotpotQA-**Q** | Given a multi-hop question, retrieve documents that can help answer the question. |
| MSMARCO-**Q** | Given a web search query, retrieve relevant passages that answer the query. |
| NFCorpus-**Q** | Given a question, retrieve relevant documents that best answer the question. |
| NQ-**Q** | Given a question, retrieve Wikipedia passages that answer the question. |
| QuoraRetrieval-**Q** | Given a question, retrieve questions that are semantically equivalent to the given question. |
| SCIDOCS-**Q** | Given a scientific paper title, retrieve paper abstracts that are cited by the given paper. |
| SciFact-**Q** | Given a scientific claim, retrieve documents that support or refute the claim. |
| Touche2020-**Q** | Given a question, retrieve detailed and persuasive arguments that answer the question. |
| TRECCOVID-**Q** | Given a query on COVID-19, retrieve documents that answer the query. |
| STS12 | Retrieve semantically similar text. |
| STS13 | Retrieve semantically similar text. |
| STS14 | Retrieve semantically similar text. |
| STS15 | Retrieve semantically similar text. |
| STS16 | Retrieve semantically similar text. |
| STS17 | Retrieve semantically similar text. |
| STS22 | Retrieve semantically similar text. |
| BIOSSES | Retrieve semantically similar text. |
| SICK-R | Retrieve semantically similar text. |
| STSBenchmark | Retrieve semantically similar text. |
| SummEval | Given a news summary, retrieve other semantically similar summaries. |
| CQADupstackTexRetrieval-**Q** | Represent the title of a user question to find a duplicate user question title with body from the Tex StackExchange forum. |
| CQADupstackTexRetrieval-**C** | Represent the question title with body posted by a user to find a duplicate user question title from the Tex StackExchange forum. |
| CQADupstackWebmastersRetrieval-**Q** | Represent the title of a user question to find a duplicate user question title with body from the Webmasters StackExchange forum. |
| CQADupstackWebmastersRetrieval-**C** | Represent the question title with body posted by a user to find a duplicate user question title from the Webmasters StackExchange forum. |
| CQADupstackEnglishRetrieval-**Q** | Represent the title of a user question to find a duplicate user question title with body from the English StackExchange forum. |
| CQADupstackEnglishRetrieval-**C** | Represent the question title with body posted by a user to find a duplicate user question title from the English StackExchange forum. |
| CQADupstackGamingRetrieval-**Q** | Represent the title of a user question to find a duplicate user question title with body from the Gaming StackExchange forum. |
| CQADupstackGamingRetrieval-**C** | Represent the question title with body posted by a user to find a duplicate user question title from the Gaming StackExchange forum. |
| CQADupstackGisRetrieval-**Q** | Represent the title of a user question to find a duplicate user question title with body from the Gis StackExchange forum. |
| CQADupstackGisRetrieval-**C** | Represent the question title with body posted by a user to find a duplicate user question title from the Gis StackExchange forum. |
| CQADupstackUnixRetrieval-**Q** | Represent the title of a user question to find a duplicate user question title with body from the Unix StackExchange forum. |
| CQADupstackUnixRetrieval-**C** | Represent the question title with body posted by a user to find a duplicate user question title from the Unix StackExchange forum. |
| CQADupstackMathematicaRetrieval-**Q** | Represent the title of a user question to find a duplicate user question title with body from the Mathematica StackExchange forum. |
| CQADupstackMathematicaRetrieval-**C** | Represent the question title with body posted by a user to find a duplicate user question title from the Mathematica StackExchange forum. |

**Table 11 – continued from previous page**

| Dataset | Instruction |
|---|---|
| CQADupstackStatsRetrieval-**Q** | Represent the title of a user question to find a duplicate user question title with body from the Stats StackExchange forum. |
| CQADupstackStatsRetrieval-**C** | Represent the question title with body posted by a user to find a duplicate user question title from the Stats StackExchange forum. |
| CQADupstackPhysicsRetrieval-**Q** | Represent the title of a user question to find a duplicate user question title with body from the Physics StackExchange forum. |
| CQADupstackPhysicsRetrieval-**C** | Represent the question title with body posted by a user to find a duplicate user question title from the Physics StackExchange forum. |
| CQADupstackProgrammersRetrieval-**Q** | Represent the title of a user question to find a duplicate user question title with body from the Programmers StackExchange forum. |
| CQADupstackProgrammersRetrieval-**C** | Represent the question title with body posted by a user to find a duplicate user question title from the Programmers StackExchange forum. |
| CQADupstackAndroidRetrieval-**Q** | Represent the title of a user question to find a duplicate user question title with body from the Android StackExchange forum. |
| CQADupstackAndroidRetrieval-**C** | Represent the question title with body posted by a user to find a duplicate user question title from the Android StackExchange forum. |
| CQADupstackWordpressRetrieval-**Q** | Represent the title of a user question to find a duplicate user question title with body from the Wordpress StackExchange forum. |
| CQADupstackWordpressRetrieval-**C** | Represent the question title with body posted by a user to find a duplicate user question title from the Wordpress StackExchange forum. |

Table 12: Instructions for two Instruction Following datasets used in **Inbedder** paper.

| Dataset | Instruction |
|---|---|
| IntentEmotion (Intent) | Represent the intent of this text. |
| IntentEmotion (Emotion) | Represent the emotion of this text. |
| NYTClustering (Location) | Represent the text based on where the news happen. |
| NYTClustering (Topic) | Represent the text based on the main news category. |

