# OpenReview forum: "Let LLMs Speak Embedding Languages: Generative Text Embeddings via Iterative Contrastive Refinement"
_ICLR.cc/2026/Conference — ICLR 2026 Poster_

### Official Review · Reviewer_Ja2S · 2025-10-21

**Soundness:** 4
**Presentation:** 4
**Contribution:** 3
**Rating:** 8
**Confidence:** 4

**Summary:**

This paper raises a problem of existing LLM-based embeddings: their overlooking of LLMs' generalization capabilities in embedding learning. To address this issue, this paper proposes a novel framework that employs autoregressive generation to iteratively refine semantic representations, thereby enhancing embedding learning. Extensive experimental results demonstrate the effectiveness of the proposed approach.

**Strengths:**

- The proposed approach is novel and interesting. It effectively exploits the generative capacity of LLMs for embedding learning.
- The soft token generation mechanism is particularly elegant, enabling the iterative refinement process to remain differentiable while capturing richer semantic information from generated tokens.
- The experimental results demonstrate competitive performance compared to strong baselines across diverse evaluation tasks, validating the effectiveness of the proposed approach.
- The analysis of iteration steps (number of generated tokens) provides valuable insights into the trade-off between computational cost and performance, offering practical guidance.

**Weaknesses:**

- While the method incurs higher computational costs than existing LLM embeddings, the KV cache can alleviate the efficiency issue.
- The paper does not provide code
- It would be better to also discuss the limitations
- The ablation study could be improved: While soft token generation is interesting and beneficial for optimization, the paper lacks a comparison between (a) directly encoding the concatenation of input and generated text, and (c) the proposed soft token approach.

**Questions:**

**Questions:**

- Could prompt engineering help steer LLMs to generate more useful tokens and potentially reduce the required number of iteration steps? It would be valuable to investigate whether task-specific or constrained prompts that guide the topic or style of generated tokens could improve both efficiency and effectiveness.
- How to set the soft token number properly? especially when using it in real-world applications.


**Suggestions:**
- The claim that "LLM-based embeddings typically adopt an encoder-only paradigm" may cause confusion, as many LLM-based embedding methods still employ LLMs in an autoregressive manner, thus maintaining a decoder-only architecture.
- Correct quotation marks to proper LaTeX formatting: "GP" → ``GP'', "TTS" → ``TTS''

---

> ### Author Response · Authors · 2025-11-19
>
> We sincerely thank the reviewer for the thorough and constructive feedback. We have found the comments very helpful in improving our paper. Below, we address each point in detail:
>
> **W1: Clarification on training cost**
>
> We appreciate the reviewer’s comment. We agree that autoregressive refinement naturally introduces overhead. However, as discussed in Appendix E, this could be substantially mitigated to nearly baseline levels (approximately 1.0–1.1× compared to Causal-EOS) through the use of KV caching technique. We will clarify this more explicitly in the main text and emphasize that the practical fine-tuning and inference cost remain close to standard LLM-based embedding models.
>
> **W2: Code availability**
>
> We acknowledge the reviewer’s concern regarding code availability. Our statement in the Reproducibility section indicates that the full codebase will be released upon publication.
> We also note that the paper provides sufficient implementation detail in the appendix to allow reproduction in the meantime. Specifically, Appendix C includes full hyperparameter configurations; Appendix D provides detailed pseudo-code for the entire generation and refinement process.
>
> **W3: Discussion of limitations**
>
> We appreciate the reviewer’s suggestion to discuss limitations. One limitation is that the number of generated refinement tokens is fixed rather than chosen adaptively; developing a mechanism to adjust this based on input complexity or compute budget would make the model more efficient. A second limitation is that the model could produce less meaningful or noisy refinement tokens; incorporating constrained decoding or token-level regularization could help improve output quality. We will include both points in the final version and consider them promising directions for future work.
>
> **W4: Concerns for ablation study**
>
> We thank the reviewer for this suggestion. We believe the experiment requested by the reviewer (specifically, (a) directly encoding the concatenation of the input and generated text) **corresponds to our "Two-Stage Generative Embedding" baseline**. As detailed in Appendix G, this baseline first uses a LLM to produce textual augmentation, which is then concatenated with the input and re-encoded. As shown in **Table 2**, our proposed GIRCSE method consistently outperforms this two-stage baseline. We hope this addresses the reviewer's concern.
>
>
> **Q1: Using prompt engineering to guide token generation**
>
> We thank the reviewer for raising this interesting direction. We agree that incorporating prompt engineering or prompt-optimization techniques may help steer the model toward generating more useful refinement tokens and could further improve performance. While our current framework does not rely on prompts to guide the generative refinement process, we acknowledge that better prompt design might yield more semantically aligned outputs. We view integrating prompt engineering or automated prompt optimization into GIRCSE as valuable future work.
>
> **Q2: How to set the soft token number properly**
>
> As discussed in Section 5.1, the number of refinement tokens (K) presents a trade-off between performance and computational cost. We found that generating more refinement steps (a higher K) typically improves quality up to a point, relative to the compute budget. Based on our experiments, we recommend using K=5 for low-latency applications and K=10–20 for high-accuracy settings.
>
> **S1: Clarification on "encoder-only" terminology**
>
> We thank the reviewer for pointing out this potential confusion. Our use of "encoder-only" was intended to contrast with methods that do not use the generative power of a decoder for the final embedding (e.g., Causal-EOS). We agree the phrasing is imprecise, as our method utilizes a decoder-only LLM architecture. In the revised manuscript, we have updated the wording to clearly describe these methods as single-pass feature extractors, avoiding any architectural confusion.
>
>
> **S2: LaTeX formatting**
>
> We thank the reviewer for identifying this formatting issue. All quotation marks in the manuscript have been corrected to proper LaTeX typographic quotes in the revised version.

---

> > ### Comment · Reviewer_Ja2S · 2025-11-21
> > **Reply to Author**
> >
> > Dear authors,
> >
> > Thank you for your prompt response. My concerns have been adequately addressed, and I will maintain my original evaluation. Please ensure that the improvements and clarifications discussed in your rebuttal are included in the camera-ready version.
> >
> > Best,
> > Ja2S

---

### Official Review · Reviewer_VYNC · 2025-10-26

**Soundness:** 2
**Presentation:** 2
**Contribution:** 3
**Rating:** 6
**Confidence:** 3

**Summary:**

This paper proposes a generative framework that lets LLMs “speak embedding languages” instead of acting as static encoders. The key idea is to iteratively generate soft tokens that refine sentence representations under a new Iterative Contrastive Refinement (ICR) objective.

**Strengths:**

1. The core idea is creative and well-motivated
2. The paper is clearly written and supported by both quantitative and qualitative analyses.
3. provides a neat way to connect generation and contrastive learning.

**Weaknesses:**

1. The framework fixes the number of refinement steps (K) during training and inference; an adaptive stopping criterion based on semantic convergence could make the process more efficient and principled.
2. While the paper claims that soft tokens capture latent semantics, there is no direct evidence or visualization showing how these tokens differ structurally from standard hidden states.
3. It is unclear whether gradient flow through the softmax-weighted soft tokens remains stable in multi-step generation. Flat or peaked distributions could cause weak or unstable gradients, potentially limiting refinement quality.

**Questions:**

1. Given that the number of refinement steps (K) is fixed, have the authors considered adaptive stopping criteria based on semantic saturation?
2. What evidence supports the claim that soft tokens capture latent semantics beyond surface-level similarity? Could the authors visualize or cluster the soft-token space?
3. Is gradient flow stable through the soft token generation? Since each soft token is a weighted sum over all vocabulary embeddings, gradients must pass through the softmax distribution. In multi-step generation, could this cause gradients to become unstable or too weak (for example, when the softmax becomes too flat or too sharp)?

---

> ### Author Response · Authors · 2025-11-19
>
> We sincerely thank the reviewer for their insightful technical questions and constructive feedback. We address each point below.
>
>  **W1, Q1: Adaptive stopping criterion**
>
> We appreciate the reviewer’s suggestion regarding an adaptive stopping criterion based on semantic saturation. We agree that such a mechanism is a valuable direction for extending GIRCSE. One natural approach is to compute the cosine similarity between the embeddings generated at steps $k$ and $k+1$, and to stop refinement once the similarity exceeds a predefined threshold. Another promising direction is to decode an EOS token similarly to standard language modeling, although this would require additional supervision or training heuristics. We are currently evaluating cosine-based saturation rules for inference and will include these results in the latter response.
>
> **W2, Q2: Do soft tokens capture latent semantics?**
>
> We thank the reviewer for raising this question. To clarify, we do not claim that individual soft tokens inherently encode latent semantics. Instead, it is the full GIRCSE generative embedding framework (especially the iterative contrastive refinement) that enables richer semantic extraction. We support this with three lines of evidence:
>
> 1. **Performance gains over fair baselines.** As shown in Table 2, GIRCSE consistently outperforms fair baselines on both MTEB and instruction-following tasks. This suggests that the embeddings produced by GIRCSE encode more semantic information than those from standard embedding approaches.
>
> 2. **Benefit of generating additional refinement tokens.** As discussed in Section 5.1, performance improves monotonically when more refinement tokens are generated. This trend indicates that the iterative refinement mechanism incrementally extracts and infers richer semantic information.
>
> 3. **Qualitative evidence from generated tokens.**  Given the above-mentioned quantitative results, we further present the qualitative experiment in Section 5.2. As shown in Table 4, GIRCSE is able to infer meaningful, semantically related tokens beyond the original input text.
>
> While it is challenging to interpret whether a model does capture latent semantics than the others, we examine this in both qualitative and quantitative ways. We hope this clarifies the concern.
>
> **W3, Q3: Gradient stability during soft-token generation**
>
> We appreciate the reviewer’s concern regarding potential instability arising from generating soft tokens. In practice, we do not observe any training instability. To address this question more directly, we have uploaded a revised version including training curves in Appendix H. As shown in Figures 4 and Figure 5, the training loss is smooth and stable, and the gradient norm converges normally throughout training. We hope this additional evidence alleviates the reviewer’s concern.

---

### Official Review · Reviewer_gt8m · 2025-10-27

**Soundness:** 2
**Presentation:** 3
**Contribution:** 2
**Rating:** 4
**Confidence:** 4

**Summary:**

This paper presents a novel framework to train that enables generative embedding models. Unlike naive next token prediction which leads to nondifferentiability, it incorporates soft-token generation which can be seen as the weighted average of possible next tokens. Furthermore, these soft-tokens are trained with contrastive objective which is regularized by the their new objective which ensures that newly generated soft-tokens lead to better performance. Overall, the framework adapts existing decoder models to generative text embedding models which eventually could lead to test-time scaling behaviour observed in recent reasoning models.

**Strengths:**

* The framework is novel, and idea of test-time scaling is relevant and interesting given the recent inference-time scaling efforts.
* Performance is robust across different datasets and models.
* Paper is well-written and easy to follow.

**Weaknesses:**

* A significant test-time scaling [methodology](https://jina.ai/news/scaling-test-time-compute-for-embedding-models/) for embedding model is not cited.
* Test-time scaling claims are ambitious, gains are marginal or non-existent. Even though methodology is interesting, I dont think generating more tokens leads to similar observations made in o1/s1 style reasoning works. Scores barely change, and decrease in the causal-eos is expected since they are not trained to generate new tokens. Comparison with a model like GritLM would be more accurate.

* 5-20 tokens are really low for test-time scaling, generating hundreds of tokens would be more interesting to draw any conclusions. Moreover, model is limited to 5 token generation during training. This is quite conservative and reduces the scope of paper.

*  Instruction following setup is not ideal. They are often used to measure models' generative performance. Popular benchmarks are MMLU, GSM8K, Alpaca, Human eval etc. Current evaluation setup is not diverse enough to capture instruction following or any other generative task performance. GritLM's setup is much more preferred and should be used in my opinion.









* There is a typo in line 153, it should be decoder instead of encoder.

**Questions:**

1) What is the performance of standart embeddings after your training methodology?
2) Why do you only train with soft tokens? Can't you use the standart embeddings as an additional objective?
3) You train with query-answer pair, have you tried using wikipedia 1 million datasets from SimCSE to train SimCSE/LLM2Vec style where we rely on augmentations?
4) What is the number of non-soft tokens, because even generating 20 soft tokens is small given the large document or query lengths?

---

> ### Author Response · Authors · 2025-11-19
>
> We sincerely thank the reviewer for their detailed feedback. We address each point below, including new experimental results that we believe will clarify our contributions and address the reviewer's concerns.
>
>
> **W1: Missing citation of test-time scaling on embedding model.**
>
> We thank the reviewer for highlighting this related work. The provided link refers to a blog post. As it is not a formal, peer-reviewed paper or a pre-print (e.g., on arXiv), we were unable to cite it per academic standards. We will gladly cite it if a citable version becomes available before publication.
>
> **W2. Ambitious test-time scaling claims & marginal improvements**
>
> We appreciate the reviewer's feedback and would like to clarify three points:
>
> 1. **Clarification on Contribution:** We would like to clarify that the **primary contribution of our work is the introduction of a novel embedding paradigm**, not an exploration of test-time scaling itself. We present the scaling behavior as an interesting and emergent property of our GIRCSE method, rather than its central claim.
>
> 2. **On Marginal Improvements:** We agree that the averaged performance in Figure 2 may appear incremental. The averaged result is intended to be useful for a concise visualization. However, we also note that this could overlook the significant gains on individual datasets. To provide clearer results, we present a detailed breakdown of performance on the STS dataset below.
>     The results demonstrate that generating more tokens **leads to substantial improvements on several key datasets**. For instance, on STS13, performance increases from 0.6311 (1 token) to 0.7168 (20 tokens), a 13.58% relative improvement. This confirms that the test-time refinement is not marginal.
>
> |   # Generated Tokens     | BIOSSES         | SICK-R          | STS12           | STS13            | STS14           | STS15           | STSBenchmark    | STS17           | STS22.v2        |
> |:------ |:--------------- |:--------------- |:--------------- |:---------------- |:--------------- |:--------------- |:--------------- |:--------------- |:--------------- |
> | **1**  | 0.7878          | 0.7379          | 0.7205          | 0.6311           | 0.6884          | 0.8352          | 0.8039          | 0.7751          | 0.6689          |
> | **3**  | 0.7855 (-0.29%) | 0.7312 (-0.91%) | 0.721 (+0.07%)  | 0.6456 (+2.30%)  | 0.7022 (+2.00%) | 0.8377 (+0.30%) | 0.8106 (+0.83%) | 0.7782 (+0.40%) | 0.6764 (+1.12%) |
> | **5**  | 0.7813 (-0.83%) | 0.7294 (-1.15%) | 0.7241 (+0.50%) | 0.665 (+5.37%)   | 0.7159 (+3.99%) | 0.8406 (+0.65%) | 0.8179 (+1.74%) | 0.7839 (+1.14%) | 0.6802 (+1.69%) |
> | **10** | 0.7839 (-0.50%) | 0.7276 (-1.40%) | 0.7288 (+1.15%) | 0.6962 (+10.32%) | 0.735 (+6.77%)  | 0.8438 (+1.03%) | 0.8278 (+2.97%) | 0.7951 (+2.58%) | 0.6847 (+2.36%) |
> | **15** | 0.7853 (-0.32%) | 0.7269 (-1.49%) | 0.73 (+1.32%)   | 0.7101 (+12.52%) | 0.7423 (+7.83%) | 0.8444 (+1.10%) | 0.8303 (+3.28%) | 0.7985 (+3.02%) | 0.6887 (+2.96%) |
> | **20** | 0.7865 (-0.17%) | 0.7265 (-1.54%) | 0.7304 (+1.37%) | 0.7168 (+13.58%) | 0.7457 (+8.32%) | 0.8446 (+1.13%) | 0.8315 (+3.43%) | 0.801 (+3.34%)  | 0.6904 (+3.21%) |
>
> 3. **Test-time Scaling with GRITLM model:** To directly address the reviewer's suggestion, we ran the same test-time scaling experiment on GritLM. The results (table below) show that, similar to the Causal-EOS baseline, generating more tokens **does not** improve GritLM's performance and in fact degrade it. This new result supports our finding that the ability to improve via test-time generation is a **unique and effective property of our GIRCSE method**, not a general property of all embedding models.
>
> | # Generated Tokens | 0 | 1 | 3 | 5 | 10 | 15 | 20 |
> |:---- |:--:|:--:|:--:|:--:|:---:|:---:|:---:|
> | Classification | 81.25 | 69.45 | 70.58 | 70.99 | 70.98 | 70.98 | 70.70 |
> | Clustering | 50.82 | 45.25 | 46.00 | 45.95 | 46.00 | 46.84 | 46.71 |
> | STS | 83.03 | 69.32 | 71.46 | 73.19 | 75.72 | 76.96 | 77.14 |

---

> ### Author Response · Authors · 2025-11-19
>
> **W3. limitation of TTS claim due to low token uses**
>
> We would like to clarify two points regarding the number of generated tokens:
>
> **Clarification on generating hundreds of tokens.**
> We agree that exploring a larger token regime is an interesting direction. However, our experiments in Figure 2 show that **performance effectively converges around 20 tokens**, with diminishing returns beyond that point. Therefore, we believe generating hundreds of tokens would offer negligible gains while significantly increasing inference cost.
>
> **Number of tokens during training.**
> To address the reviewer's concern about using only 5 tokens during training, we conducted experiments training GIRCSE using QWEN0.5B backbone with 3, 7, and 9 tokens. The results below show that while performance increases from 3 to 7 tokens, we observe diminishing returns, with the gain from 7 to 9 tokens being marginal (0.631465 vs. 0.631471). This supports our original choice of 5 tokens as a balanced trade-off between performance and training efficiency.
>
> | Train Tokens | Classification | Clustering | PairClass. | Reranking | Retrieval | STS | Summarization | **Average** |
> |:---|:---|:---|:---|:---|:---|:---|:---|:---|
> | **3** | 0.7593 | 0.4982 | 0.7926 | 0.4471 | 0.4803 | 0.7427 | 0.3279 | 0.6134 |
> | **5** | 0.7729 | 0.5128 | 0.8184 | 0.4572 | 0.4919 | 0.7447 | 0.3328 | **0.6246** |
> | **7** | 0.7824 | 0.5119 | 0.8157 | 0.4547 | 0.5004 | 0.7594 | 0.3380 | **0.6314** |
> | **9** | 0.7824 | 0.5119 | 0.8157 | 0.4547 | 0.5004 | 0.7594 | 0.3380 | **0.6314** |
>
>
> **W4: Concerns on Instruction-following evaluation.**
>
> We thank the reviewer for this comment. We believe there may be a misunderstanding of our model's task and output. **Our model does not generate textual answers.** Instead, it uses a generative process to **refine an embedding vector.**
>
> Benchmarks like MMLU, GSM8K, and HumanEval are designed to evaluate the quality of generated text (e.g., reasoning or coding). In contrast, our evaluation setup is designed to measure **embedding-level instruction following** (i.e., whether the embeddings correctly captures the instruction). As our model does not output text, evaluating it on generative benchmarks like MMLU would be inappropriate. We therefore respectfully maintain that our current evaluation setup is a correct and appropriate measure for the specific task we are addressing.
>
> **W5: Typo in line 153**
>
> Thank you for pointing out this typo. We have corrected “encoder” to “decoder” in the revised manuscript.

---

> ### Author Response · Authors · 2025-11-19
>
> **Q1. What is the performance of standard embeddings**
>
> We appreciate the reviewer's question and evaluate the performance of the standard embedding (i.e., the last token embedding of the input text) on the MTEB benchmark. The results are presented in the following table. We also include the original results of GIRCSE for reference. The results show that the standard embedding still maintains reasonable embedding quality while leveraging more generation tokens (our default setting) still achieves better performance.
>
> | type | Classification | Clustering | PairClassification | Reranking | Retrieval | STS | Summarization | Mean |
> |:---|:---|:---|:---|:---|:---|:---|:---|:---|
> | **GIRCSE (Standard Embedding)** | 0.8436 | 0.5428 | 0.8565 | 0.4921 | 0.5524 | 0.7398 | 0.3631 | 0.6632 |
> | **GIRCSE (Paper Results)** | 0.8533 | 0.5626 | 0.8618 | 0.4888 | 0.5710 | 0.7637 | 0.3356 | 0.6783 |
>
> **Q2. Why train only with soft tokens? Why not include standard embeddings as an auxiliary objective?**
>
> Our design choice is directly motivated by our goal: training the LLM to **generate a sequence of soft tokens that are then pooled to form a high-quality embedding.** Because the final embedding is derived only from these generated tokens, we focus the training objective purely on them. The standard input embeddings (e.g., the last token) are not used in the final pooling operation, so we exclude them from the objective to force the model to learn this generative refinement process.
>
> **Q3. Have you tried SimCSE-style augmentation using Wikipedia 1M?**
> We followed the training methodology of recent SOTA LLM-based embedding models (e.g., E5-Mistral [1], BGE-en-ICL [2], GRITLM [3]), which have shown that high-quality supervised pairs are more effective than unsupervised augmentations. While training with Wikipedia-style data is an interesting direction, our primary goal was to demonstrate the effectiveness of our GIRCSE paradigm using the current best-practice training data for a fair comparison with other leading models.
>
> **Q4. What is the number of non-soft tokens? Are 20 soft tokens small relative to long documents?**
>
> We would like to clarify a key aspect of our model: the **generated soft tokens are not intended to match the length of the input document.** Instead, they represent a sequence of refinement steps that are pooled to create a single, fixed-size embedding. As shown in our responses to W2 and W3, performance converges at around 20 tokens. Generating more tokens yields negligible gains while increasing inference cost. Additionally, this approach is consistent with (and an expansion of) other SOTA models. For example, models like E5-Mistral use only a single token (i.e., the `[EOS]` token) to represent the entire document. Therefore, we believe 20 tokens are not small in this context; they are a sufficient number of refinement steps to achieve a high-quality final embedding.
>
>
> ### **References**
> [1] Wang, Liang, et al. "Improving text embeddings with large language models." Proceedings of the 62nd Annual Meeting of the Association for Computational Linguistics (Volume 1: Long Papers). 2024.
>
> [2] Li, Chaofan, et al. "Making Text Embedders Few-Shot Learners." The Thirteenth International Conference on Learning Representations.
>
> [3] Muennighoff, Niklas, et al. "Generative representational instruction tuning." The Thirteenth International Conference on Learning Representations. 2024.

---

> ### Comment · Reviewer_gt8m · 2025-11-25
>
> * I dont think a citable version will come but you could still cite the blogpost anyway since many of the works from industry is in the form of blogposts.
>
> *  STS13 improvement is promising whereas, gains are not general and consistent. Have you seen a similar improvement in other tasks such as retrieval or clustering because STS is known to not to correlate with real-world task performance.
>
> * My GritLM discussion is misunderstood but it is not significant anyway.
>
> * Additional experiments are appreciated but diminishing results imply that TTS is limited because we are putting everything in a vector. Any directions to avoid this potential bottleneck?
>
> * My remark regarding Wikipedia is essentially doing contrastive learning with unsupervised examples, which is often done before pairwise contrastive learning. Since Wiki-1m is small, I would really love to see the unsupervised performance of the gircse.
>
> * My last question about the performance of the method in full training instead of lora. Have you done any experiment on smaller models, if so would you share them?
>
> * Please don't use screenshot pictures for figure 4 and 5, they look blurry and unprofessional. You could create a better quality image by pulling your data using wandb API.  If you dont know how to do it, ask chatgpt.
>
> * I would be happy to raise my score if unsupervised and lora performance is also added. You dont have to finish them till end of the rebuttal, but I believe they will strengthen the work.

---

> ### Author Response · Authors · 2025-12-03
>
> We sincerely thank the reviewer for their constructive feedback and the time taken to review our work. We appreciate the opportunity to strengthen the paper with the suggested experiments. We address the specific comments below:
>
> 1. **Citation of Industry Blog Posts**
> Thank you for this suggestion. We agree that acknowledging relevant work is important for a complete literature review. We will consider adding the citation to the specific blog post in the camera ready version.
>
> 2. **Additional test-time scaling experiment on retrieval tasks**
> We agree with the reviewer’s observation that STS performance does not always correlate with real-world tasks like retrieval. To address this, we evaluated our method on the NanoBEIR benchmark. The results demonstrate that Test-Time Scaling (TTS) remains effective in retrieval settings. As shown in the table below, increasing compute consistently improves performance, achieving substantial gains of up to **+4.66% on ArguAna** and **+4.26% on SciFact** when scaling to 20 steps.
>
> | | ArguAna | FiQA2018 | HotpotQA | MSMARCO | NQ | SciFact | Touche2020 |
> | :--- | :--- | :--- | :--- | :--- | :--- | :--- | :--- |
> | **1** | 0.6719 | 0.5874 | 0.8772 | 0.6282 | 0.7738 | 0.7911 | 0.4584 |
> | **3** | 0.6882 (+2.43%) | 0.5986 (+1.91%) | 0.8813 (+0.47%) | 0.6366 (+1.34%) | 0.7807 (+0.89%) | 0.8002 (+1.15%) | 0.4599 (+0.33%) |
> | **5** | 0.6890 (+2.55%) | 0.6038 (+2.79%) | 0.8863 (+1.04%) | 0.6414 (+2.10%) | 0.7907 (+2.18%) | 0.8007 (+1.21%) | 0.4670 (+1.88%) |
> | **10** | 0.6868 (+2.22%) | 0.6077 (+3.46%) | 0.8808 (+0.41%) | 0.6470 (+2.99%) | 0.7839 (+1.31%) | 0.8248 (+4.26%) | 0.4687 (+2.25%) |
> | **15** | 0.6917 (+2.95%) | 0.6048 (+2.96%) | 0.8872 (+1.14%) | 0.6398 (+1.85%) | 0.7857 (+1.54%) | 0.8240 (+4.16%) | 0.4670 (+1.88%) |
> | **20** | 0.7032 (+4.66%) | 0.6079 (+3.49%) | 0.8808 (+0.41%) | 0.6403 (+1.93%) | 0.7811 (+0.94%) | 0.8164 (+3.20%) | 0.4608 (+0.52%) |
>
> 3. **Clarification on GritLM**
> We are glad that our previous response helped clarify the discussion regarding GritLM and resolved the concern.
>
> 4. **Discussion on mitigating diminishing results with more generated tokens**
> We thank the reviewer for this insightful observation regarding the information bottleneck. We agree that compressing an extensive chain of generated tokens into a single fixed-size vector limits capacity, which likely contributes to the diminishing returns observed after 20 tokens. To address this, we propose two concrete directions for future work to alleviate this bottleneck:
>
> * **Multi-Vector Framework:** One potential direction is extending GIRCSE to a multi-vector architecture (similar to ColBERT [1]). By preserving the generated soft tokens for late interaction (MaxSim) rather than pooling them immediately, the model avoids the single-vector bottleneck. This would likely require adding a MaxSim-based auxiliary loss during training.
> * **Enhanced Single-Vector Pooling:** Alternatively, to maintain the efficiency of a single-vector representation, the diminishing returns might stem from the generated tokens being semantically redundant. Future work could explore diversity regularization during training to encourage the model to capture distinct semantic information per token. Additionally, replacing simple average pooling with a learnable Latent Attention Layer (as utilized in NV-Embed [2]) could help the model compress this information more effectively into the final embedding.
>
> We believe these training-time optimizations are key to resolving the saturation issue, allowing the model to fully benefit from increased generation steps at inference.
>
>
> 5. **Training GIRCSE with unsupervised contrastive learning**
> We agree that evaluating unsupervised performance offers a clearer view of the method’s capability. We ran GIRCSE on Wiki-1M, but the limited rebuttal time prevented us from obtaining reliable results. Specifically, we observed that the training loss rapidly decreased to zero. The same phenomenon also happens when training the fair baseline under the same setting. This suggests that the issue stems from the current configuration of the unsupervised training pipeline (e.g., the lack of effective hard negatives or specific hyperparameter tuing on this dataset) rather than a flaw in the GIRCSE architecture itself. While we could not resolve this within the rebuttal window, we believe GIRCSE has the potential to work in this setting with proper tuning.
>
> ### **References**
>
> [1] Khattab, Omar, and Matei Zaharia. "Colbert: Efficient and effective passage search via contextualized late interaction over bert." Proceedings of the 43rd International ACM SIGIR conference on research and development in Information Retrieval. 2020.
>
> [2] Lee, Chankyu, et al. "NV-Embed: Improved Techniques for Training LLMs as Generalist Embedding Models." The Thirteenth International Conference on Learning Representations.

---

> > ### Author Response · Authors · 2025-12-03
> >
> > 6. **Full Fine-tune vs. LoRA on smaller models**
> > We conducted a comparative experiment between full fine-tuning and LoRA using the QWEN2.5-0.5B model. Due to time constraints during the rebuttal period, these models were trained on a sampled subset of 50k examples. We plan to scale this up to 200k samples for the camera-ready version to verify the consistency of these findings. As shown in the table below, while full fine-tuning yields a slight performance advantage due to the update of all parameters, LoRA achieves highly competitive performance (within ~1% on average compared to full fine-tuning). We hope this result validates our design choice to focus on LoRA for the main experiments to balance performance with computational accessibility.
> >
> > | Type | Classification | Clustering | PairClassification | Reranking | Retrieval | STS | Summarization | Average |
> > | :--- | :--- | :--- | :--- | :--- | :--- | :--- | :--- | :--- |
> > | **GIRCSE-LoRA** | 0.7421 | 0.4727 | 0.7800 | 0.4584 | 0.4092 | 0.7724 | 0.3083 | 0.5933 |
> > | **GIRCSE-Full-FT**| 0.7393 | 0.4724 | 0.7902 | 0.4523 | 0.4616 | 0.7589 | 0.2851 | 0.6024 |
> >
> > 7. **Figure Quality (Figures 4 & 5)**
> > We appreciate the advice regarding figure quality. We have regenerated Figures 4 and 5 using the raw data from our experiment logs rather than screenshots to ensure they are clear and professional. These have been updated in the revised PDF.
> >
> > We hope that the additional experiment and response addresses your concerns.

---

### Official Review · Reviewer_HLGU · 2025-10-31

**Soundness:** 4
**Presentation:** 2
**Contribution:** 3
**Rating:** 8
**Confidence:** 4

**Summary:**

The paper "Let LLMs Speak Embedding Languages..." aims to address the problem that algorithms generate representations from text using LLMs are usually static, in that they just make one fixed embedding from e.g. the last layer, last token embeddings. To address this they propose a method to iteratively refine the embeddings over several steps. The algorithm auto-regressively generates so-called "soft-tokens" which should give the model a richer ability to refine the representations. This has the drawback of requiring a fine-tuning of the language model but they show that you only need to use a limited amount of data to still achieve decent performance improvements. The primary empirical results show for popular models, (mistral, qwen) that this algorithm can reliably improve the learned representations.

**Strengths:**

I think the primary strengths of this work are as follows:
1. Originality. The iterative algorithm and soft-token idea is well founded and intuitive, and to the best of my knowledge is novel at least in this use case.
2. Quality. This paper uses a very reasonable suite of evaluations and tests appropriate models that people are typically using in practice.
3. Clarity. I think the empirical results are presented well.
4. Significance. This algorithm is general, and I think it can be applied to many different architectures. The ability to scale this method in terms of the number of time steps in a test-time compute fashion is quite novel to the field of representation learning, and to my knowledge has not been as successful as this paper demonstrates in the past.

**Weaknesses:**

I would happily raise my score on the "presentation" category if these are addressed:
1. I thought that the presentation of the GIRCSE algorithm was not clear in the main text in Section 3.2 / 3.3. I did however find that the "Algorithm 1" pseudo-code description made this much more obvious what is going on. I would recommend moving this to the main text if space permits.
2. Figure 1 is not very clear, it looks more like what would be expected for a poster than a figure introducing the algorithm.

**Questions:**

1. Is the fine-tuning stage of this algorithm more computationally expensive then regular SFT on the same .2B tokens?

---

> ### Author Response · Authors · 2025-11-19
>
> We sincerely thank the reviewer for their constructive feedback and the willingness to raise the presentation score. We hope the following revisions will fully address the remaining concerns.
>
> **W1: Clarity of Section 3.2 / 3.3 and Algorithm Description**
>
> We thank the reviewer for this constructive suggestion. We agree that Algorithm 1 (currently in the appendix) provides the clearest description of the GIRCSE embedding paradigm. We were unable to include it in the main text due to the strict 9-page submission limit.
> We will move Algorithm 1 into Section 3 for the camera-ready version, which permits an additional page. We will also revise the surrounding text in Sections 3.2 and 3.3 to integrate it smoothly. We are confident this change will greatly improve the paper's clarity, and we appreciate the reviewer for identifying this path to improvement.
>
> **W2: Clarity of Figure 1**
>
> We thank the reviewer for raising this concern. We would like to clarify that Figure 1 functions primarily as a high-level conceptual illustration rather than a detailed algorithmic diagram. Since we have now moved Algorithm 1 into the main paper (see reply of W1), the manuscript already provides a clear and precise description of the full procedure. In this context, we believe that Figure 1 can remain a conceptual overview without duplicating the algorithmic details.
>
> **Q1: Computational Cost of Fine-tuning**
>
> GIRCSE does introduce a slightly higher computational overhead during fine-tuning compared to a standard SFT. As detailed in Appendix E/F, this is substantially mitigated by using KV caching. During training, we generate 5 soft tokens. With KV caching, the effective FLOPs are only 1.0–1.1× that of a standard SFT on the same dataset.

---

### Author Response · Authors · 2025-12-03

We thank the Area Chair and all reviewers for their constructive feedback. We are encouraged that three out of four reviewers (HLGU, Ja2S, VYNC) voted for acceptance (Scores: 8, 8, 6), acknowledging the originality of our generative embedding paradigm, the elegance of the soft-token mechanism, and the solid empirical results.

Below, we summarize our response to the primary concerns, particularly addressing the divergence in **Reviewer gt8m**’s assessment.

* **Marginal improvement for test-time scaling**: We provided a detailed breakdown on STS and retrieval tasks showing that averaged results masked significant gains on many tasks. For instance, scaling to 20 tokens yields a 13.58% relative improvement on STS13.
* **Evaluation on generative benchmarks (e.g., MMLU):** The reviewer suggested evaluating on MMLU/GSM8K. We believe this is a misunderstanding and clarified that our model is **an embedding model, not a text generation model**. Our output is a vector, not text. Therefore, standard embedding benchmarks (e.g., MTEB) are the correct evaluation metric, while generative benchmarks like MMLU are inapplicable.
* **Generating hundreds of tokens**: Our experiments in Figure 2 show that performance effectively converges around 20 tokens, with diminishing returns beyond that point. Therefore, we believe generating hundreds of tokens would offer negligible gains while significantly increasing inference cost.

We have incorporated the valuable suggestions from the positive reviewers to further strengthen the paper:

* Algorithm Visibility (HLGU): We will move Algorithm 1 from the appendix to the main text (Section 3) to improve the clarity of GIRCSE's presentation.
* Gradient Stability (VYNC): We added training dynamics plots (Appendix H) demonstrating that the loss curves are smooth and gradient norms are stable. This address the concerns about the stability of soft-token generation.
* Clarification on ablation study and limitations (Ja2S): We clarified the two-stage baseline comparison and explicitly discussed the limitations in the revised text.

**Conclusion**

We believe the rebuttal has fully addressed the technical concerns regarding test-time scaling raised by Reviewer gt8m. Given the strong consensus among three reviewers regarding the novelty and effectiveness of our approach, we are confident that this work makes a valuable contribution to the field of LLM-based representation learning.

---

### Meta-Review · Area_Chair_dzrH · 2026-01-07

**Summary:**

This paper proposes GIRCSE, a generative embedding method for decoder-only LLMs. Instead of taking a single static embedding, it autoregressively generates differentiable “soft tokens” and pools their hidden states into the final vector, trained with an iterative contrastive objective to encourage step-by-step refinement. Reviews are mostly positive. Strengths are clear novelty, an elegant soft-token mechanism, and strong results on standard embedding benchmarks across models. Main concerns are clarity (algorithm/figures), gradient stability, and the consistency of test-time scaling gains. The rebuttal adds stability evidence, stronger per-task results (including retrieval), and commits to clearer presentation.

**Reviewer Concerns:**

- Rebuttal addressed: algorithm/figure clarity, gradient stability in multi-step soft-token generation, and the strength/consistency of test-time scaling by adding stability plots, per-task breakdowns (including retrieval), and committing to clearer presentation.

- Still outstanding: limited evidence in unsupervised settings, saturation beyond ~20 refinement - steps (single-vector bottleneck), and incomplete large-scale comparisons (e.g., full fine-tuning vs. LoRA beyond small subsets)

**Reviewer Scores:**

- HLGU: likely would have kept a strong accept, since the authors committed to moving Algorithm 1 into the main text and clarified training cost.
- Ja2S: would have kept an accept, as they explicitly said their concerns were addressed and they would maintain the original evaluation.
- VYNC: likely would have stayed slightly above threshold, since stability evidence and discussion of adaptive stopping were added, but some broader interpretability/analysis requests remain.
- gt8m: likely would have raised slightly, given the added retrieval scaling results and extra experiments, though they still flagged outstanding requests like longer-step scaling

---

### Decision · Program_Chairs · 2026-01-26

Accept (Poster)